# Collaborative Learning via Prediction Consensus

**Dongyang Fan**[1]    **Celestine Mendler-Dünner**[2,3]    **Martin Jaggi**[1]

[1]EPFL, Switzerland
[2]Max Planck Institute for Intelligent Systems, Tübingen, Germany
[3]ELLIS Institute Tübingen, Germany
{dongyang.fan, martin.jaggi}@epfl.ch
celestine@tue.ellis.eu

## Abstract

We consider a collaborative learning setting where the goal of each agent is to improve their own model by leveraging the expertise of collaborators, in addition to their own training data. To facilitate the exchange of expertise among agents, we propose a distillation-based method leveraging shared unlabeled auxiliary data, which is pseudo-labeled by the collective. Central to our method is a trust weighting scheme that serves to adaptively weigh the influence of each collaborator on the pseudo-labels until a consensus on how to label the auxiliary data is reached. We demonstrate empirically that our collaboration scheme is able to significantly boost individual models' performance in the target domain from which the auxiliary data is sampled. At the same time, it can provably mitigate the negative impact of bad models on the collective. By design, our method adeptly accommodates heterogeneity in model architectures and substantially reduces communication overhead compared to typical collaborative learning methods.

## 1   Introduction

This work considers a decentralized learning setting where each agent has access to a labeled dataset and a local model. The agents may differ in the data distribution they have access to as well as the quality of their local models. In addition, we assume a shared unlabeled dataset $X^*$ sampled from a target distribution $\mathcal{Q}$ is available to all agents. The central question is *how can agents effectively exchange information to benefit from each other's local expertise in order to improve their predictive performance on the target domain $\mathcal{Q}$?*

Towards this goal, our work takes inspiration from social science on how a panel of human experts collaborate on a task. Humans typically engage in discourse to exchange information, they share their opinions, and based on how much they trust their peers, each individual will then adjust their subjective belief towards the opinion of peers. When repeated, this process gives rise to a dynamic process of consensus finding, as formalized by DeGroot [1]. Central to the consensus mechanism of DeGroot is the concept of *trust*. It determines how much individual agents influence each other's opinion, and thus the influence of each agent on the final consensus.

Our proposed algorithm mimics this consensus-finding mechanism in the context of collaborative learning, inspired by [2]. In particular, our consensus procedure is aimed at how to label the shared dataset $X^*$. Therefore, we carefully design a strategy by which each agent determines the trust towards others, given its local information, to optimally leverage each agent's expertise to collectively pseudo-label $X^*$. This mechanism of knowledge distillation is then combined with techniques from self-training [3] in order to transfer the shared knowledge from the pseudo-labels into the local models in an iterative fashion.

Code available at `https://github.com/fan1dy/collaboration-consensus`

37th Conference on Neural Information Processing Systems (NeurIPS 2023).

Crucial to our approach is that instead of aiming for a consensus on model parameters, such as typically done in federated learning, we leverage the abundance of unlabeled data to enforce consensus in *prediction space* over $X^*$. A key benefit of information exchange in prediction space is a reduction in communication complexity and the ability to control privacy leakage, but also the ability to seamlessly cope with both data heterogeneity and model heterogeneity.

**Problem setup.** We consider a collaborative learning setup with $N$ agents. Each agent $i \in [N]$ holds local training data, sampled from a local data distribution $\mathcal{P}_i$. We denote the local training data as $(X_i, y_i)$, where the matrix $X_i$ composes of $n_i$ local datapoints and $y_i$ denotes the vector of corresponding labels. The number $n_i$ needs not to be the same across agents. In addition, we assume agents have access to a shared unlabeled dataset $X^*$ sampled from a target distribution $\mathcal{Q}$. We use $n^*$ to denote the number of datapoints in $X^*$. The ultimate success measure we consider is each agent's prediction performance on the target distribution $\mathcal{Q}$. We work under the assumption that sharing of raw labeled data is not desirable due to privacy concerns, data ownership, or storage constraints and we wish to keep communication at a minimum to avoid overheads. Our setting recovers the goal of both decentralized and federated learning, with or without personalization [4], when $\mathcal{Q}$ is defined as the prediction task on the mixture of all local data distributions. In addition, we allow agents to differ in their model architectures. We do not require a coordination server but assume agents have all-to-all communication available.

**Contributions.** The key contributions of our work can be summarized in the following aspects:

1. We propose a novel collaborative learning algorithm based on prediction-consensus, which effectively addresses statistical and model heterogeneity in the learning process, and provably mitigates the negative impact of low-quality participating agents.
2. Our algorithm is able to significantly reduce communication overhead and privacy-sensitive information sharing in comparison to other collaborative learning methods while achieving superior empirical performance.
3. We show theoretically that consensus can be reached via our algorithm and justify the conditions for good consensus to be achieved.

## 2   Related work

In *federated learning* a central server coordinates local updates toward learning a global model. Local nodes upload gradients or model parameters, instead of data itself, to maintain a certain level of privacy. McMahan et al. [5] describe the classic FedAvg algorithm. Follow-up works mainly focus on addressing challenges from non-i.i.d. local data [6, 7, 8] and robustness towards Byzantine attacks. Apart from communicating gradients or model parameters, several works discuss alternatives to allow for heterogeneous model architectures. These methods are based on variants of model distillation [9, 10], reaching an agreement in the representations space [11] or output space [12]. Similar to our work, both assume access to a shared unlabeled dataset. However, we go beyond naive averaging to determine agreement to account for heterogeneity in model or data quality.

In contrast to federated learning, the *fully decentralized learning* setting does not assume the existence of a central server. Instead, decentralized schemes such as gossip averaging are used to aggregate local information across agents [13, 14]. Despite the lack of a global state, such methods can provably converge to the desired global solution, leading to a gradual consensus among individual models [4]. In this context, Dandi et al. [15], Le Bars et al. [16] optimizes the communication topology to adapt to data heterogeneity but do not offer any collaborator selection mechanisms. Bellet et al. [13] allows personalized models on each agent, but assumes prior information about task-relatedness, as opposed to learned selection. *Gossip algorithms* typically assume a fixed gossip mixing matrix given by e.g. physical connections of nodes [17, 18, 19]. These approaches fail to consider data-dependent communication as with task similarities and node qualities. Several recent works have addressed this issue by proposing alternative methods that consider these factors. Notably, Li et al. [20] directly optimizes the mixing weights by minimizing the local validation loss per node, which requires labeled validation sets. Sui et al. [21] uses the E-step of EM algorithm to estimate the importance of other agents to one specific agent $i$, by evaluating the accuracy of other agents' models on the local data of agent $i$. This way of trust computation does not allow the algorithm to be applied to target distributions that differ from the local distribution, differentiating it from our work. Moreover, for

both Li et al. [20], Sui et al. [21] the aggregation is performed in the gradient space, therefore not allowing heterogeneous models.

Our work relates to *semi-supervised learning* as it involves partially unlabeled data. Most relevant are self-training methods [22] that first train a model using labeled data, then use the trained model to give pseudo-labels to unlabeled data. The pseudo-labels can further be fed back to the training loop to attain a better model. Wei et al. [23] shows that under expansion and separation assumptions, self-training with input consistency regularization can achieve high accuracy with respect to ground-truth labels. When more than one learner is involved, co-training [24] appears as an extension to self-training, benefiting from the knowledge of learners from independent views in labeling a set of unlabeled data. Diao et al. [25] incorporates SSL into federated learning. In a setting where agents are with unlabeled data and the center server is with labeled data, experimental studies demonstrate that the performance of a labeled server is significantly improved with unlabeled clients. Farina [26] presented a collective learning framework for distributed SSL, where they combine predictions on a shared dataset via weights evaluated from local models' performances on local validation datasets. While their algorithm bears similarities to ours, it is important to note that it is exclusively tailored to scenarios in which the target domain matches the global distribution. In a similar spirit, we want to leverage unlabeled data in a fully decentralized setting.

Finally, Mendler-Dünner et al. [2] have previously formalized collective prediction as a dynamic *consensus finding procedure*. They demonstrated that such an approach can lead to significant gains over naive model averaging. We extend their approach from test-time prediction to collaborative model training.

# 3 Method description

Our proposed method is designed to take advantage of shared unlabeled data in the context of collaborative learning through knowledge distillation. Therefore, it emulates human opinion dynamics to collectively pseudo-label the shared auxiliary data. These labels are then incorporated in the local model update steps towards collectively improving the performance on the data distribution from which the shared data is sampled.

## 3.1 Collective pseudo-labeling

To describe the pseudo labeling step, let us use $f_{\boldsymbol{\theta}_i}$ to denote the local model of agent $i \in [N]$ parameterized by $\boldsymbol{\theta}_i$. We write $\hat{\boldsymbol{y}}_i = f_{\boldsymbol{\theta}_i}(\boldsymbol{X}^*)$ to denote the predictions of agent $i$ on the auxiliary data $\boldsymbol{X}^*$. Agents share these predictions with their peers. Naturally, the individual models may differ in these predictions and it is a priori unclear which model is most accurate, as ground truth labels of the auxiliary data are not available. To combine the predictions into pseudo-labels for $\boldsymbol{X}^*$, each agent locally decides how to weigh other agents' predictions by estimating their respective expertise on the target task. We refer to these weights as trust scores and we use $w_{ij}$ to denote the trust of agent $i$ towards the predictions of agent $j$. It's worth noting that the trust between agents is not necessarily mutual, i.e., can be asymmetrical: agent $i$ can trust agent $j$ without agent $j$ necessarily trusting agent $i$ back. We use $\boldsymbol{W}$ to denote the matrix of trust scores. Given the trust scores, agent $i$ uses the following pseudo labels for the auxiliary data:

$$\boldsymbol{\psi}_i = \sum_j w_{ij}\hat{\boldsymbol{y}}_j. \tag{1}$$

Trust scores are determined locally by each agent based on query access to other agents' predictions and they are refined iteratively throughout training as models are being updated. The adaptive weight computation will be detailed in Section 4.

## 3.2 Collaborative learning from pseudo labels

In the second step, the proxy labels for the auxiliary data are used to augment local model training. Therefore, in each step, the local optimization problem is augmented by a disagreement loss, and the new objective is given by

$$\mathcal{L}(f_{\boldsymbol{\theta}_i}(\boldsymbol{X_i}), \boldsymbol{y_i}) + \lambda \mathrm{dist}(f_{\boldsymbol{\theta}_i}(\boldsymbol{X}^*), \boldsymbol{\psi}_i) \tag{2}$$

---

**Algorithm 1** Pseudo code of our proposed algorithm

---

**Input:** For each agent $i \in [N]$ we are given a local model $\boldsymbol{\theta}_i^{(0)}$, a labeled local dataset $(\boldsymbol{X_i}, \boldsymbol{y_i})$, and unlabeled shared data $\boldsymbol{X}^*$.

**for** $t = 1, ..., T$ **do**

    Each node $i \in [N]$ broadcasts their soft labels $\hat{\boldsymbol{y}}_i^{(t-1)} = f_{\boldsymbol{\theta}_i^{(t-1)}}(\boldsymbol{X}^*)$ to all other nodes

    **in parallel for** each agent $i$ **do**

        • Calculate pairwise trust score $w_{ij}^{(t)} (j \in [N])$, based on the received soft decisions using methods provided in Section 4

        • Get pseudo-labels on $\boldsymbol{X}^*$ from collaborators: $\boldsymbol{\psi}_i^{(t)} = \sum_j w_{ij}^{(t)} \hat{\boldsymbol{y}}_j^{(t-1)}$

        • Do local training with collaborative disagreement loss

$$\boldsymbol{\theta}_i^{(t)} \in \arg\min_{\boldsymbol{\theta}} \ \mathcal{L}(f_{\boldsymbol{\theta}}(\boldsymbol{X_i}), \boldsymbol{y_i}) + \lambda \text{dist}(f_{\boldsymbol{\theta}}(\boldsymbol{X}^*), \boldsymbol{\psi}_i^{(t)}) \tag{3}$$

**end for**

---

where $f_{\boldsymbol{\theta}_i}(\boldsymbol{X})$ denotes the vector of agent $i$'s predictions on the dataset $\boldsymbol{X}$, $\mathcal{L}$ is the local training loss and $\text{dist}(\cdot)$ is a disagreement measure. We choose $l_2$ distance for the disagreement measure in the regression case and cross-entropy for the classification case. $\lambda > 0$ is a trade-off hyperparameter that weighs the local loss and the cost of disagreement. This objective adheres to a conventional semi-supervised learning approach, however, we generate pseudo-labels in a trust-based collective manner.

To iteratively refine the local models in the spirit of self-training, the pseudo labeling step and the local training step are performed in an alternating fashion as described in Algorithm 1. Starting from pre-trained models $\boldsymbol{\theta}_i^{(0)}$, in each round $t \in \{1, .., T\}$ model predictions on the auxiliary data are shared and then each agent aggregates them into a set of pseudo labels to augment local data and perform an update step.

Our algorithm is motivated conceptually by co-training [24] where it was demonstrated that unlabeled data can be used to augment labeled data to boost model performance. Moreover, learning from collective pseudo-labels offers several additional benefits. Firstly, aggregation is performed in the prediction space, eliminating the need for all agents to have identical model architectures. Secondly, the communication cost of transmitting predictions is significantly lower than that of sharing model weights, and the same pseudo-labels $\boldsymbol{\psi}_i$ can be reused for multiple local epochs to further reduce the communication burden.

### 3.3 Convergence analysis

We study under what conditions Algorithm 1 will reach a consensus among agents on how to label the auxiliary data. For the analysis, we focus on the over-parameterized regime [1] and we make the following assumption on the local data distributions:

**Assumption 1.** *There is no concept shift between the local data distributions and the target domain $\mathcal{Q}$ from which the shared data is sampled, i.e., $\mathcal{P}_i(Y|X{=}x) = \mathcal{Q}(Y|X{=}x)$ for all $i \in [N]$.*

Together with over-parameterization the assumption implies that the minimizer of the objective specified in (2) can always reach zero loss. Further, this allows us to model the update of agents' predictions on $\boldsymbol{X}^*$ as a Markov process where the state transition matrix corresponds to the trust matrix $\boldsymbol{W}^{(t)}$. Therefore, it is convenient to write the update of the predictions on $\boldsymbol{X}^*$ performed by the algorithm in matrix form, as $\boldsymbol{\Psi}^{(t)} = [\hat{\boldsymbol{y}}_1^{(t)}, .., \hat{\boldsymbol{y}}_N^{(t)}]$. Adopting this notation we have for $t \geq 1$

$$\boldsymbol{\Psi}^{(t)} = \boldsymbol{W}^{(t)} \boldsymbol{\Psi}^{(t-1)} = \boldsymbol{W}^{(t)} \boldsymbol{W}^{(t-1)} ... \boldsymbol{W}^{(1)} \boldsymbol{\Psi}^{(0)} . \tag{4}$$

The following result provides sufficient conditions under which consensus will be reached by our algorithm.

---

[1]We say a model is over-parameterized if its training error can reach zero. Over-parameterization is a reasonable assumption in the deep learning regime.

**Theorem 1** (Consensus on predictions). *Assume all agents' models are over-parameterized and the data distributions satisfy Assumption 1. Then, for $t \to \infty$ Algorithm 1 converges to a consensus among the local models on the predictions on $\boldsymbol{X}^*$, that is,*

$$\boldsymbol{\psi}_i^{(t)} = \boldsymbol{\psi}_j^{(t)} \quad \forall i \neq j, \tag{5}$$

*as long as $\boldsymbol{W}^{(t)}$ is row-stochastic and positive for any $t \geq 0$.*

The proof is given in Appendix B and the main insight is that as long as $\boldsymbol{W}$ is row-stochastic and positive (that is $\sum_j w_{ij} = 1$ for any row $i$, and $w_{ij} > 0$), the product of any $\boldsymbol{W}^{(t)}$'s is stochastic, irreducible, and aperiodic, and this leads to the differences between rows in $\boldsymbol{\Psi}^{(t)}$ vanishing in time. Together with no concept shift, over-parameterization is important to guarantee that models can fit the consensus predictions on the unlabeled data while at the same time minimizing local losses. In contrast, in the under-parameterized setting, consensus and local loss minimization cannot necessarily be achieved at the same time.

### 3.4   Information sharing in prediction space

A key feature of our method is that agents do not share model parameters, but they communicate by exchanging prediction queries. If Algorithm 1 achieves a consensus this means that agents arrive at solutions where they agree on predictions on $\boldsymbol{X}^*$, but this does not imply that they have learned the same model, or that they agree on predictions outside $\boldsymbol{X}^*$. To illustrate this, we provide a simple example where local data are generated using cubic regression with additive i.i.d. noise in the output, as shown in Fig. 1. In this example, uniform weights are the ideal solution. We apply this optimal uniform trust, as the main purpose here is to illustrate the difference between information sharing in prediction space and parameter space. Each agent fits a polynomial regression of degree 4, which leads to *over-parameterization* of the model to fit the data. Full details of our example are given in Appendix A.1. We refer to the work of [2] for a similar setting with under-parameterized models. Here we note the most interesting observations in the over-parameterized regime.

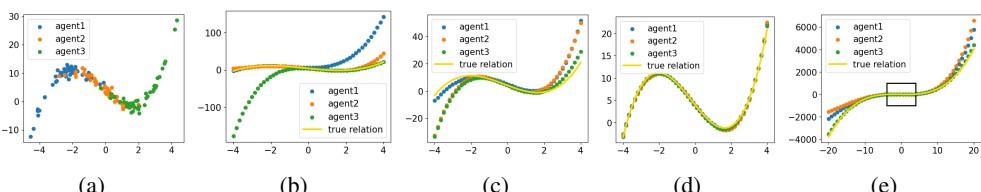

|     |     |     |     |     |
| --- | --- | --- | --- | --- |
| (a) | (b) | (c) | (d) | (e) |

Figure 1: Local data distributions are shown in (a), and the initial fit on local data is shown in (b). (c) and (d) are predictions on $\boldsymbol{X}^*$ after 5 rounds and 20 rounds of our algorithm update respectively. (e) is the comparison of model fits in a larger range ((d) is zoom-in of the rectangular area of (e))

First, we observe that for $T \geq 20$ the three agents reach a consensus on the predictions of $\boldsymbol{X}^*$. However, the model parameters are not the same across the agents, as depicted in the rightmost panel. Further, considering the properties of the algorithm across rounds and the predictions in different regions of the input space in more detail, the following desirable behaviors are observed:

a) In the region where agent $i$ has more data, it fits the local data more accurately and it moves pseudo-labels closer to its own predictions.

b) In the region where agent $i$ has no or little data, agent $i$ only updates its model parameters to fit the pseudo-labels.

c) When local loss minimization and prediction consensus can be achieved at the same time, agents can arrive at models with a perfect agreement in the target prediction space.

## 4   Design of trust weights

In Section 3.3 we have shown that our algorithm is guaranteed to reach consensus on $\boldsymbol{X}^*$ under weak assumptions on the trust matrix $\boldsymbol{W}^{(t)}$. In this section, we discuss how to design $\boldsymbol{W}^{(t)}$ to encourage that the achieved consensus leads to a high-quality labeling of $\boldsymbol{X}^*$.

Therefore, we focus on multi-class classification. We let $f_{\boldsymbol{\theta}_i}(\boldsymbol{x})$ denote the class probabilities obtained using model $\boldsymbol{\theta}_i$ for a datapoint $\boldsymbol{x}$. We choose the cross-entropy measure $\mathcal{H}(\cdot, \cdot)$ to define the agreement loss function $\mathrm{dist}(\cdot, \cdot)$ in (3). If $\boldsymbol{X} = [\boldsymbol{x}_1, .., \boldsymbol{x}_n]^\top$, then $f_{\boldsymbol{\theta}}(\boldsymbol{X}) = [f_{\boldsymbol{\theta}}(\boldsymbol{x}_1), .., f_{\boldsymbol{\theta}}(\boldsymbol{x}_n)]^\top \in \mathbb{R}^{n \times C}$, where $n$ is the number of samples in $\boldsymbol{X}$ and $C$ is the number of classes.

### 4.1 Trust evaluation through self-confidence

The quality of the local models could differ due to various factors, such as the amount of labeled data available during training, due to the expressivity of the local model, the training algorithm, or due to the relevance of the local data for the target task of labeling $\mathcal{Q}$. Thus, a desirable property of the consensus solution is that malicious agents, or agents with low-quality models contribute less to the pseudo-labeling than agents with better models.

Hadjicostis and Dominguez-Garcia [27] differentiate between malicious and non-malicious agents and they discuss the concept of trustworthy consensus, where only non-malicious agents contribute to the consensus. In contrast to prior work, we do not aim for trustworthy agents to contribute equally. Instead, we specifically want consensus to come from potentially *unequal* contribution of all agents, weighted according to their relevance. We allow for the trust matrix to be asymmetric. All agents determine trust from information given locally to the respective agent, which differs across agents. Central to any such strategy is that the capabilities of models on $\mathcal{Q}$ can be estimated appropriately. In the following, we discuss a strategy of how to determine trust from local data and prediction queries to other models.

As no label information on $\boldsymbol{X}^*$ is available to evaluate trust, it is natural to use agents' own predictions on $\boldsymbol{X}^*$ as a local reference point. Then, each agent distributes their trust towards other agents based on the alignment of their predictions. We use weighted pairwise cosine similarity as a measure of alignment which motivates the following trust weight calculation:

$$w_{ij}^{(t)} = \frac{\gamma_{ij}^{(t)}}{\sum_j \gamma_{ij}^{(t)}} \quad \text{with} \quad \gamma_{ij}^{(t)} = \frac{1}{n^*} \sum_{\boldsymbol{x} \in \boldsymbol{X}^*} \beta_i^{(t)}(\boldsymbol{x}) \frac{\left\langle f_{\boldsymbol{\theta}_i^{(t-1)}}(\boldsymbol{x}), f_{\boldsymbol{\theta}_j^{(t-1)}}(\boldsymbol{x}) \right\rangle}{\|f_{\boldsymbol{\theta}_i^{(t-1)}}(\boldsymbol{x})\|_2 \|f_{\boldsymbol{\theta}_j^{(t-1)}}(\boldsymbol{x})\|_2}. \tag{6}$$

The inclusion of the weighting factor $\beta_i^{(t)}(\boldsymbol{x})$ and how to choose it will be discussed in Section 4.2.

**Self-confident trust.** Naturally, pairwise cosine similarity leads to a trust matrix that has diagonal entries being the highest value among each row. We call this property *self-confident*, as each agent trusts itself the most. We now demonstrate that this property is not particularly restrictive. Even if constraining trust matrix to be self-confident, it is still possible to design such a matrix that facilitates any consensus. The proof is given in Appendix D.

**Proposition 2.** *For any given consensus distribution $\boldsymbol{\pi}$, it is always possible to find a trust matrix $\boldsymbol{W}$ that leads to it, which is both row stochastic and self-confident.*

A second nice property our trust calculation has is that for an appropriate choice of $\beta_i^{(t)}$ the proposed calculation of trust scores in (6) leads to scores that become more evenly distributed over time as agents gradually reach consensus.

**Claim 3.** *Given Assumption 1 holds and that all agents are over-parameterized. Assume $\beta_i^{(t)}$ is chosen such that the trust matrix $\boldsymbol{W}^{(t)}$ is row-stochastic and positive for all $t \geq 0$. Then, for the trust calculation in (6), we have $\boldsymbol{W}^{(t)}$ loses self-confidence over time and finally converges to a uniform matrix:*

$$tr(\boldsymbol{W}^{(t)}) \leq tr(\boldsymbol{W}^{(t-1)}) \quad \text{and} \quad \boldsymbol{W}^{(t)} \overset{t \to \infty}{\Longrightarrow} \mathbf{1}\mathbf{1}^\top \frac{1}{N} \tag{7}$$

The proof is provided in Appendix C. This claim characterizes the behavior of our dynamic trust scheme: while initially all agents distribute trust towards helpful collaborators and try to achieve a consensus on $\boldsymbol{X}^*$. Once consensus is reached, we will have $\hat{\boldsymbol{y}}_i^{(t)} = \hat{\boldsymbol{y}}_j^{(t)}$ for any $i, j$, and $\boldsymbol{W}^{(t)}$ will become a matrix with uniform weights. This means no individual agent has increasingly high weight or the ability to manipulate the labeling.

Now that we know that a self-confident matrix can lead to the desired consensus, what other properties should our trust matrix have?

## 4.2 Robustness to low-quality nodes

If agents possess low-quality local data, we aim to minimize their influence on the labeling of the auxiliary data throughout the algorithm. Proposition 4 gives a sufficient condition for such a desired consensus: if there exists only one node with low-quality data, as long as it receives the lowest trust from other regular nodes, and the sum of trust it receives is the smallest compared to the others, it will remain to have lowest importance in the consensus.

**Proposition 4.** *Given Assumption 1, the trust matrix is row stochastic and positive, and all agents hold over-parameterized models. Let $b$ be the only node with low-quality data and $\tau$ be the timestep that consensus is reached. If the following desirable properties hold for $t < \tau$:*

 *i)  $b$ receives the lowest trust from others than itself, i.e., $w_{jb}^{(t)} = \min_i w_{ji}^{(t)}$ for $j \neq b$.*

 *ii)  $b$-th column has the lowest column sum: $\sum_j w_{jb}^{(t)} < \min_{i \neq b} \sum_j w_{ji}^{(t)}$.*

*Then node $b$ will have the lowest importance in the consensus.*

The proof is given in Appendix E, where we also provide desired properties in the presence of multiple nodes with low-quality data, under some extra assumptions. Proposition 4 emphasizes the desired trust weights during training *before* consensus is reached.

When nodes with weak model architectures (such as under-parameterized models) are involved, achieving consensus is not assured. If such a consensus solution does exist, it will be constrained by the underfitting of weak nodes. Consequently, this solution would not serve as a stationary solution concerning the local training loss of a strong node. Nevertheless, we conjecture that these desired properties can still enhance training by mitigating the impact of the weak nodes.

## 4.3 Confidence weighting

In the following paragraph, we discuss the choice of the weights $\beta_i^{(t)}$ in (6). Specifically, we incorporate confidence weighting into the pairwise cosine similarity calculation to emulate the construction of a transition matrix based on a known consensus distribution.

Let us start by outlining an idealized trust calculation that effectively down-weighs agents with low quality data. We first construct an intermediate transition matrix $\boldsymbol{\Phi}$ from pairwise cosine similarities of the agents' predictions on $\boldsymbol{X}^*$ (with row normalization). For the low-quality node $b$, we will have $\phi_{jb}$ being the lowest value in the $j$-th row, for any $j \neq b$. According to Proposition 4, in order to have low importance of low-quality workers in the consensus, we need to set the overall trust that $b$ receives to be the lowest among all the nodes. To achieve this, we need to assign the trust of regular workers towards the low-quality workers to a very small value, as it is difficult to alter self-confidence. If the consensus importance weight is known, one can easily calculate the corresponding trust matrix

$$w_{jb} = \phi_{jb} \min \left( 1, \frac{\pi(b)}{\pi(j)} \frac{\phi_{bj}}{\phi_{jb}} \right),$$

which is a classical result from Metropolis chains [28] (also see Appendix D). We will have $w_{jb} < \phi_{jb}$ for $j \neq b$, as $\frac{\pi(b)}{\pi(j)}$ should be sufficiently small.

**Practical scheme.** Since the consensus importance weight is unknown, we cannot attain the ideal trust matrix. Therefore, we propose an alternative weighting scheme that achieves similar effects: we up-weight the similarity in the region where agent $j$ has more confidence, i.e., where agent $j$'s class probability assignments have lower entropy. By doing this, we encourage that the trust weights become more concentrated on themselves and helpful workers, and less concentrated on low-quality workers. We incorporate this into the trust weight calculation (6) by choosing

$$\beta_i^{(t)}(\boldsymbol{x}) = \frac{1}{\mathcal{H}(f_{\boldsymbol{\theta}_i^{(t-1)}}(\boldsymbol{x}))}$$

where $\mathcal{H}$ denotes the entropy. We offer further intuition as well as justification of this weighting scheme in Appendix F. Moreover, we empirically demonstrate how our choice of trust matrix leads to a low column sum for bad nodes in Section 5.2.

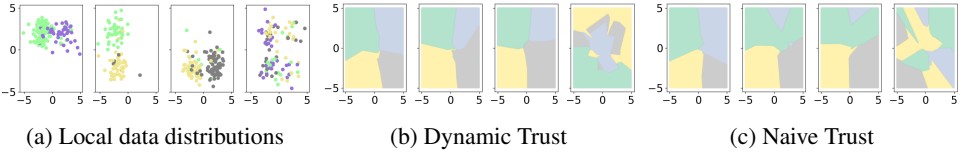

|(a) Local data distributions|(b) Dynamic Trust|(c) Naive Trust|

Figure 2: Decision boundary comparison between our dynamic trust update and naive trust update

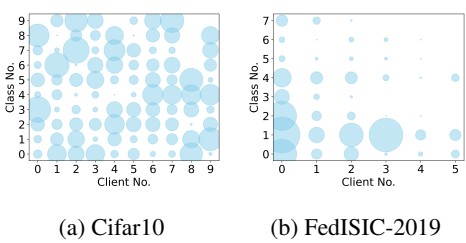

(a) Cifar10      (b) FedISIC-2019

Figure 3: Class distributions among clients

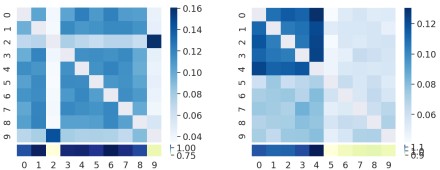

Figure 4: Learned trust matrix during training with diagonal entries masked and column sum reported in the lowest bar. (left) Agents 2&9 have bad data. (right) Agents 5-9 have weak models.

## 5 Experiments

We start with a synthetic example to visualize the decision boundary achieved by our algorithm and then demonstrate its performance on real data in a heterogeneous collaborative learning setting.

### 5.1 Decision boundary visualization

Four classes are generated via multivariate Gaussian following $P^c \sim \mathcal{N}(\boldsymbol{\mu}_c, \boldsymbol{\Sigma})$, where $\boldsymbol{\mu}_0 = (-2, 2)^\top$, $\boldsymbol{\mu}_1 = (2, 2)^\top$, $\boldsymbol{\mu}_2 = (-2, -2)^\top$, $\boldsymbol{\mu}_3 = (2, -2)^\top$. $\boldsymbol{\Sigma} = \boldsymbol{I}_{2\times2}$. Four clients have local data sampled from a mixture of $P^c$'s. For clients 0-3, we randomly flip 10% of the labels, and for client 3, we flip all labels. The unlabeled data $\boldsymbol{X}^*$ are sampled equally from $P^c$'s. The data distribution is shown in Fig. 2a. The base model used in each node is a multi-layer perceptron of 3 layers with 5, 10, and 4 neurons respectively. We now compare Algorithm 1 with dynamic trust weight to with naive trust weight. When a client with low-quality data is involved, i.e. client 3 in the toy example, our trust update scheme gives a better decision boundary to good agents after collaboration, as blind trust towards low-quality clients will impair the effectiveness of pseudo labeling.

### 5.2 Deep learning experiments

**Datasets and model architectures.** We consider a more challenging setting, where local data distributions are non-i.i.d. Two different statistical heterogeneities are considered: (1) *Synthetic heterogeneity*. We utilize the classic Cifar10 and Cifar100 datasets [29] and create 10 clients from each dataset. To distribute classes among clients, we use a Dirichlet distribution[2] with $\alpha = 1$. Unless specified otherwise, we employ ResNet20 [30] without pretraining. (2) *Real-world data heterogeneity*. A real-world dermoscopic lesion image dataset from the ISIC 2019 challenge [31, 32, 33] is included here. The same client splits are used as in [34], based on the imaging acquisition system employed in six different hospitals. The dataset includes eight classes of lesions to classify, with the class distribution among the clients displayed in Fig. 3b. Following [34], we choose pretrained EfficientNet [35] as the base model, and use *balanced accuracy* as the evaluation metric. For every dataset, we construct $\boldsymbol{X}^*$ from equally contributed samples by every agent.

**Comparison against baseline methods.** We compare our methods with several baseline methods, including FedAvg [5], FedProx [7], SCAFFOLD [8] (SCA), FedDyn [36], local training without collaboration (LT), and training with naive trust (Naive). Note with naive trust we are realizing soft majority voting, which represents the baseline method proposed from [12]. We adhere to the same

---

[2]The Dirichlet distributed samples are constructed using the codes from `https://github.com/TsingZ0/PFL-Non-IID`

|  |  | FedAvg | FedProx | SCA | FedDyn | LT | Naive | Ours-S | Ours-D |
|---|---|---|---|---|---|---|---|---|---|
| Regular | Cifar10 | 0.542 | 0.517 | 0.578 | 0.578 | 0.475 | **0.618** | 0.604 | 0.612 |
|  | Cifar100 | 0.261 | 0.240 | 0.317 | 0.310 | 0.178 | 0.311 | **0.319** | 0.308 |
|  | Fed-ISIC | 0.279 | 0.261 | 0.213 | 0.243 | 0.248 | 0.290 | **0.302** | 0.291 |
| Low-Quality Data | Cifar10 | 0.541 | 0.530 | 0.570 | 0.575 | 0.470 | 0.596 | 0.605 | **0.608** |
|  | Cifar100 | 0.254 | 0.240 | 0.289 | **0.308** | 0.171 | 0.285 | 0.300 | 0.306 |
|  | Fed-ISIC | 0.229 | 0.242 | 0.221 | 0.243 | 0.217 | 0.247 | 0.249 | **0.269** |

Table 1: Our methods compare to baseline methods. **Blue** denotes the algorithm with top 1 accuracy and green denotes the method with 2nd best accuracy. "Ours - S" denotes the static version where the trust score is kept constant after first-time calculation (after 5 rounds of local training) and "Ours - D" denotes the dynamic version where the trust score is updated per global round.

architecture setting, where the standard federated learning algorithms can be applied. To initiate the process, we allow each client to perform local training for 5 global rounds, with the objective of obtaining a sufficiently refined model that can be used for trust evaluation. From the 6th training round, the clients start collaboration. Over a total of 50 global rounds, each consisting of 5 local epochs, we report the averaged accuracy results from three repeated experiments in Table 1. The evaluation metric is calculated on the dataset $X^*$. $\lambda$ is fixed as 0.5 in all experiments.

When all nodes share the same data quality and degree of statistical heterogeneity (denoted by "regular" in the table), our methods align closely with consensus through naive averaging, which is optimal in this case. When all nodes share the same degree of statistical heterogeneity but differ in data quality, exemplified by randomly selecting two nodes (indexed as 2 and 9) for a complete flip of local training labels, our dynamic trust update shows better overall performances[3], proving the effectiveness of our approach in limiting the detrimental influences from nodes with low-quality data. We further plotted out the learned trust matrix in the dynamic update mode during one of the middle training rounds in the left plot of Fig. 4. Clearly, our algorithm is able to give low trust weights to the nodes with low-quality data, and the 2nd and 9th columns have the lowest column sum.

**Adaptability to varying model architectures.** We allocate a more expressive model architecture to the first half of the nodes and a less expressive one to the other half. The former comprises ResNet20 and EfficientNet, which were the models of choice in the previous experiments. For the latter, we employ a linear model (i.e., one-layer fully connected neural network) with a flattened image tensor as input and the output is of size equivalent to the number of classes. It is worth noting that if agents with strong and weak model architectures (as in cases of under-parameterization) coexist, consensus might not occur, as suggested by our empirical findings illustrated in Fig. 5. Nevertheless, our trust-based collaborator selection mechanism consistently outperforms local training and simple averaging. The trust weight matrix learned during Cifar100 training is depicted in the right plot of Fig. 4, revealing the presence of asymmetric trust. Specifically, the last 5 nodes exhibit a higher level of trust towards the first 5 nodes, while the opposite is not true. The trust allocation is desired in identifying the helpers. We further refer to Appendix A.2 for more empirical evidence from a toy polynomial regression example on the presence of strong and weak architectures.

**Reduced communication costs.** Gradient aggregation-based methods incur a significant communication burden proportional to the number of model parameters ($\mathcal{O}(N \times |params|)$), which is particularly heavy given the over-parameterized nature of modern deep learning.

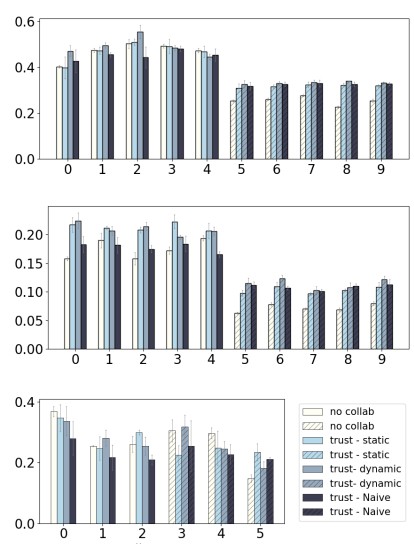

Figure 5: Target accuracy comparison with 2 different model architectures with error bars (hatch pattern denotes fully connected NN is used). From top to bottom: Cifar10, Cifar100, FedISIC

---

[3]Here we report average accuracy of regular workers, excluding workers with low-quality data

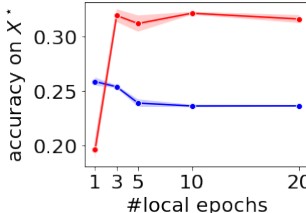 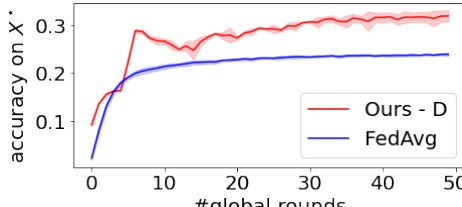

Figure 6: Algorithm performances on Cifar100 for different algorithm configurations. (left) effect of varying number of local epochs on final performance; (right) algorithm performance as a function of the number of training rounds for 5 local epochs each

In contrast to existing approaches, our proposed method significantly reduces the communication burden by enabling each node to transmit only their predictions on the shared dataset. This results in communication overhead $\mathcal{O}(N^2 \times n^\star \times C)$. It is clear that this value does not scale up with more complex models, and is much smaller than the model size. Moreover, our methods maintain their high performance even when the number of local epochs increases. On the other hand, FedAvg loses its effectiveness with less frequent synchronization, i.e. more local epochs between global aggregation rounds, as shown in the left panel of Fig. 6.

## 6 Conclusions and extensions

In the context of decentralized learning, we leverage the collective knowledge of individual nodes to improve the accuracy of predictions with respect to a target distribution. Our proposed trust update scheme, based on self-confidence, ensures robustness against nodes with low-quality data. By achieving consensus in the prediction space, our method effectively handles diverse model architectures within local clients, while maintaining a low communication overhead, thereby exhibiting important practical potential. Despite coming from a different perspective, our trust-based collaborative pseudo-labeling method may provide some inspiration in the semi-supervised learning community. Notably, our algorithm is intrinsically compatible with personalization, in terms of allowing some concept shift across clients. We leave this for future work.

**Robustness.** We have designed our algorithm with the assumption that all agents communicate *honestly*, meaning that no Byzantine workers *intentionally* provide incorrect information. Nevertheless, our method exhibits some resilience against a common Byzantine attack, known as the label flip (referred to as "low-quality workers" in our paper). For instance, even with 2 out of 10 workers having 100% flipped labels, our algorithm maintains good performance. If there are malicious workers deliberately providing incorrect information, the nodes may refuse to reach a consensus, instead of reaching a detrimental bad consensus, assuming a reasonable $\lambda$ is chosen. Consider the scenario in which a detrimental consensus is reached with malicious nodes involved; in this case, the consensus loss and local loss for regular nodes will not decrease in the same direction, making the consensus solution non-stationary. Notably, the "personal" component of our loss function adds an element of robustness against malicious nodes.

**Privacy Concerns.** While previous works show that training data can be reconstructed from model parameters [37] or gradients [38], our algorithm requires less privacy-sensitive information sharing, which is predictions on a shared dataset. While we are aware that model predictions can still leak private information on training data due to memorization [39], there is a trade-off between the gain from collaboration and the amount of information that users are willing to share. As the number of outer rounds increases, we observe a notable improvement in accuracy within the context of $\boldsymbol{X}^*$. However, this enhanced accuracy comes at the cost of disclosing more information, a relationship that is depicted in Fig. 6. An interesting extension would be to apply differential privacy to further guarantee privacy.

**Acknowledgements.** DF would like to thank Anastasia Koloskova, Felix Kuchelmeister, Matteo Pagliardini, and Nikita Doikov for helpful discussions during the project and El Mahdi Chayti for proofreading. DF acknowledges funding from EDIC fellowship from the Department of Computer Science at EPFL. CM acknowledges support from the Tübingen AI Center. This project was supported by SNSF grant 200020_200342.

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

# Appendix

## A  Polynomial regression examples

### A.1  Example in Section 3.4

The true underlying function is chosen as $f(x) = 0.5x^3 + 0.3x^2 - 5x + 4$. There are three agents in total, each of whom has 50 data points. The local data points are generated using normal distributions: $x_1 \sim \mathcal{N}(-2, 1)$, $x_2 \sim \mathcal{N}(0, 1)$ and $x_3 \sim \mathcal{N}(2, 1)$. To introduce noise in the labels, each agent adds a normally distributed error term with zero mean and unit variance, i.e. $y_i = f(x_i) + \varepsilon$ with $\varepsilon \sim \mathcal{N}(0, 1)$.

A set of 50 equally spaced data points in the range of $-4$ to $4$, denoted as $\boldsymbol{X}^*$, is used in the analysis. The algorithm is applied using fixed trust weights with 1/3 in each entry and $\lambda$ is chosen as 1.

### A.2  Example with strong and weak architectures

The true underlying function is chosen as $f(x) = 0.5x^3 + 0.3x^2 - 5x + 4$. There are four agents in total, each of whom has 50 data points. The local data points are generated using normal distributions: $x_1 \sim \mathcal{N}(-2, 1)$, $x_2 \sim \mathcal{N}(0, 1)$, $x_3 \sim \mathcal{N}(2, 1)$ and $x_4 \sim \mathcal{N}(3, 1)$. To introduce noise in the labels, each agent adds a normally distributed error term with zero mean and unit variance, i.e. $y_i = f(x_i) + \varepsilon$ with $\varepsilon \sim \mathcal{N}(0, 1)$.

A set of 50 equally spaced data points in the range of $-4$ to $6$, denoted as $\boldsymbol{X}^*$, is used in the analysis. The algorithm is applied using dynamic trust weights and $\lambda$ is chosen as 1. For the first three agents, a polynomial model with a maximum degree of four is fit, while for the fourth agent, a polynomial model with a maximum degree of one is fit, signifying a weak node.

We see that after 50 rounds of model training using our proposed algorithm with dynamic trust, agent 4's model is still underfitting due to its limited expressiveness. Agents 1-3 end up agreeing with each other and giving good predictions in the union of their local regions. While with naive trust weights, we see that the strong agents also get influenced in the region where they could perform well, as the underfitted model has a stronger impact through collective pseudo-labeling.

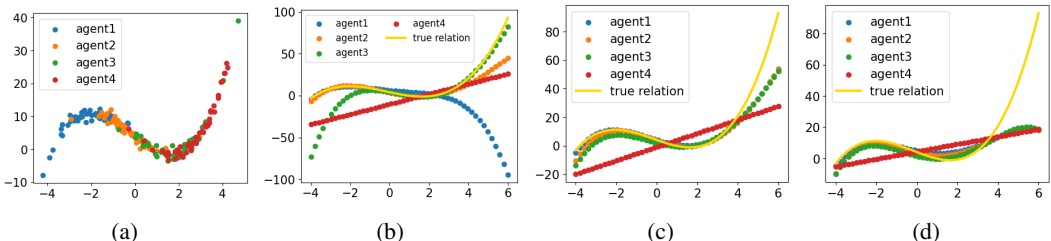

Figure 7: (a) local data distributions in each agent; (b) local model fit without collaboration; (c) model fits after 50 rounds of our algorithm with dynamic trust update; (d) model fits after 50 rounds with naive trust update

## B  Proof of Theorem 1

The proof is rooted in the results from the work of Wolfowitz [40], we recommend readers to check the original paper for more detailed references. Note, for the following texts, when we say a matrix $\boldsymbol{W}$ has certain properties, it is equivalent to saying a Markov chain induced by transition matrix $\boldsymbol{W}$ has certain properties.

**Definition A** (Irreducible Markov chains). *A Markov chain induced by transition matrix $\boldsymbol{W}$ is irreducible if for all i, j, there exists some $n$ such that $\boldsymbol{W}_{ij}^n > 0$. Equivalently, the graph corresponding to $\boldsymbol{W}$ is strongly connected.*

**Definition B** (Strongly connected graph). *A graph is said to be strongly connected if every vertex is reachable from every other vertex.*

**Definition C** (Aperiodic Markov chains). *A Markov chain induced by transition matrix $\boldsymbol{W}$ is aperiodic if every state has a self-loop. By self-loop, we mean that there is a nonzero probability of remaining in that state, i.e. $w_{ii} > 0$ for every $i$.*

**Assumption 2.** *$\boldsymbol{W}^{(t)}$'s are row-stochastic and positive , i.e. $\sum_j w_{ij} = 1$ for any row $i$, and $w_{ij} > 0$.*

**Claim 5.** *Given Assumption 2, the matrix product of any $n$ elements of $\{\boldsymbol{W}^{(t)}\}$ are SIA (SIA stands for stochastic, irreducible and aperiodic) for $n \geq 1$.*

*Proof.* According to the assumption that all $\boldsymbol{W}^{(t)}$'s are positive, and thus we have any product of $\boldsymbol{W}^{(t)}$'s being positive in each entry, which is equivalent to the graph introduced by the product being fully connected. Being fully connected implies being strongly connected. According to Definitions A B, irreducibility follows.
By the product being positive, we also have its diagonal entries being all positive. According to Definition C, aperiodicity follows.
The product of row-stochastic matrices remains row-stochastic: for $\boldsymbol{A}$ and $\boldsymbol{B}$ row stochastic, we have the product $\boldsymbol{A}\boldsymbol{B}$ remains row-stochastic.

$$\sum_j (\sum_k a_{ik} b_{kj}) = \sum_k a_{ik} (\sum_j b_{kj}) = 1, \ \forall i$$

Thus, we have any product of $\boldsymbol{W}^{(t)}$'s being irreducible, aperiodic and stochastic (SIA). $\qquad\square$

**Theorem 6** (Rewrite of Wolfowitz [40]). *Let $\boldsymbol{A}_1, ..., \boldsymbol{A}_k$ be square row stochastic matrices of the same order and any product of the $\boldsymbol{A}$'s (of whatever length) is SIA. When $k \to \infty$, the product of $\boldsymbol{A}_1, ..., \boldsymbol{A}_k$ gets reduced to a matrix with identity rows.*

Following Assumptions 12, we have $\psi^{(t)} = \boldsymbol{W}^{(t)}\psi^{(t-1)}$ holds for all $t \geq 1$. From Claim 5, we have any products of $\boldsymbol{W}^{(t)}$'s being SIA. From Theorem 6, we have the product $\boldsymbol{W}^{(t)}\boldsymbol{W}^{(t-1)} \ldots \boldsymbol{W}^{(1)}$ gets reduced to a matrix with identical rows when $t$ goes to infinity. That implies, $\psi^\infty$ has identical rows. The statement is thus proved.

## C   Proof of Claim 3

**Definition D** (Row differences). *Define how different the rows of $\boldsymbol{W}$ are by*

$$\delta(\boldsymbol{W}) = \max_j \max_{i_1, i_2} |w_{i_1 j} - w_{i_2 j}| \tag{8}$$

*For identical rows, $\delta(\boldsymbol{W}) = 0$*

**Definition E** (Scrambling matrix). *$\boldsymbol{W}$ is a scrambling matrix if*

$$\lambda(\boldsymbol{W}) := 1 - \min_{i_1, i_2} \sum_j \min(w_{i_1 j}, w_{i_2 j}) < 1 \tag{9}$$

In plain words, Definition E says that if for every pair of rows $i_1$ and $i_2$ in a matrix $\boldsymbol{W}$, there exists a column $j$ (which may depend on $i_1$ and $i_2$) such that $w_{i_1 j} > 0$ and $w_{i_2 j} > 0$, then $\boldsymbol{W}$ is a scrambling matrix. It is easy to verify that a positive matrix is always a scrambling matrix.

**Lemma 1** (Adaptation of Lemma 2 from Wolfowitz [40]). *For any $t$,*

$$\delta(\boldsymbol{W}^{(t)}\boldsymbol{W}^{(t-1)} \ldots \boldsymbol{W}^{(1)}) \leq \prod_{i=1}^{t} \lambda(\boldsymbol{W}^{(i)}) \tag{10}$$

Lemma 1 states that multiplying with scrambling matrices will make the row differences smaller. $tr(\boldsymbol{W}^{(t)}) = \sum_i w_{ii}^{(t)}$ represents the sum of self-confidences of all nodes. As every $\boldsymbol{W}^{(t)}$ is positive, we have all $\boldsymbol{W}^{(t)}$'s scrambling. Thus, the differences between rows of $\boldsymbol{W}^{(t)}\boldsymbol{W}^{(t-1)}..\boldsymbol{W}^{(1)}$ get smaller when $t$ gets bigger.

As $\boldsymbol{\psi}_i^{(t)} = \sum_j [\boldsymbol{W}^{(t)}\boldsymbol{W}^{(t-1)}..\boldsymbol{W}^{(1)}]_{ij} \boldsymbol{\psi}_j^{(t-1)}$, we have the predictions on $\boldsymbol{X}^*$ given by all nodes get similar over time. According to our calculation of $\boldsymbol{W}^{(t)}$ in Equation (6), which is based on cosine

similarity between predictions, it follows that an agent's trust towards the others gets larger over time. That is, $\sum_j w_{ij}^{(t+1)} \geq \sum_j w_{ij}^{(t)}$. Since each row sums up to 1, we have $w_{ii}^{(t+1)} \leq w_{ii}^{(t)}$, for all $i$.

According to Theorem 1, we have $\psi_i^{(t)} = \psi_j^{(t)}$ as $t \to \infty$, for any $i$ and $j$. According to the calculation of $\boldsymbol{W}$, we have $\boldsymbol{W}^{(t)}$ with equal entries when $t$ reaches infinity.

## D    Proof of Proposition 2

Recall stationary distribution ($\boldsymbol{\pi} \in \mathbb{R}^{1 \times N}$) of a Markov chain being

$$\lim_{t \to \infty} \boldsymbol{W}^{(t)} \ldots \boldsymbol{W}^{(1)} \to [\boldsymbol{\pi}^\top \ldots \boldsymbol{\pi}^\top]^\top \tag{11}$$

The proof follows from the construction of Metropolis chains given a stationary distribution. We will first give an example of how Metropolis chains work.

**Example 2** (Metropolis chains [28]). *Given stationary distribution $\boldsymbol{\pi} = [0.3, 0.3, 0.3, 0.1]$, how could we construct a transition matrix that leads to the stationary distribution?*

*Suppose $\boldsymbol{\Phi}$ is a symmetric matrix, one can construct a Metropolis chain $\boldsymbol{P}$ as follows:*

$$p(x, y) = \begin{cases} \phi(x, y) \min\left(1, \frac{\pi(y)}{\pi(x)}\right) & y \neq x \\ 1 - \sum_{z \neq x} \phi(x, z) \min\left(1, \frac{\pi(z)}{\pi(x)}\right) & y = x \end{cases} \tag{12}$$

*Choose $\boldsymbol{\Phi} = \begin{bmatrix} 1/3 & 1/4 & 1/4 & 1/6 \\ 1/4 & 1/3 & 1/4 & 1/6 \\ 1/4 & 1/4 & 1/3 & 1/6 \\ 1/6 & 1/6 & 1/6 & 1/2 \end{bmatrix}$, we could get $\boldsymbol{P} = \begin{bmatrix} 4/9 & 1/4 & 1/4 & 1/18 \\ 1/4 & 4/9 & 1/4 & 1/18 \\ 1/4 & 1/4 & 4/9 & 1/18 \\ 1/6 & 1/6 & 1/6 & 1/2 \end{bmatrix}$.*

*It can be verified that $\boldsymbol{\pi}$ is the stationary distribution of Markov chain with transition matrix $\boldsymbol{P}$. If $\boldsymbol{\Phi}$ is not symmetric, we modify $\frac{\pi(y)}{\pi(x)}$ to $\frac{\pi(y)}{\pi(x)} \frac{\phi(y,x)}{\phi(x,y)}$, and the results remain unchanged.*

Following Example 2, choose $\boldsymbol{\Phi}$ to be any self-confident doubly stochastic matrix. For all $x$, choose $\boldsymbol{P}$ as calculated from (12), we have

$$p(x, x) = 1 - \sum_{z \neq x} \phi(x, z) \min\left(1, \frac{\pi(z)}{\pi(x)}\right) \geq 1 - \sum_{z \neq x} \phi(x, z) = \phi(x, x) \tag{13}$$

we see that probability distribution among each row gets more concentrated on the diagonal entries in $\boldsymbol{P}$ than $\boldsymbol{\Phi}$. As $\boldsymbol{\Phi}$ already has high diagonal values, the claim follows.

## E    Proof of Proposition 4

Proposition 4 states sufficient conditions for $\boldsymbol{W}^{(t)}$'s to have such that a low-quality node $b$ is assigned lowest importance in $\boldsymbol{\pi}$, i.e. $\pi_b = \min_i \pi_i$. From Equation (11), $\boldsymbol{\pi}$ comes from the product of trust matrices. We start from a product of two such matrices.

**Proposition 7.** *For row-stochastic and positive matrices $\boldsymbol{A}$ and $\boldsymbol{B}$, and $\boldsymbol{C} = \boldsymbol{AB}$, if in both $\boldsymbol{A}$ and $\boldsymbol{B}$,*

    *i)  $j$-th column has the lowest column sum,*

    *ii)  $(i, j)$-th entry being the lowest value in $i$-th row for $i \neq j$*

*then we have $j$-th column remains the the lowest column sum in matrix $\boldsymbol{C}$ and $(i, j)$-th entry being the lowest value in $i$-th row of $\boldsymbol{C}$ for $i \neq j$,*

*Proof.* Let $\boldsymbol{C} = \boldsymbol{AB}$, the column sum of column $j$ of $\boldsymbol{C}$ can be expressed as:

$$\begin{aligned} \sum_i c_{ij} &= \sum_i \sum_k a_{ik} b_{kj} \\ &= \sum_k (\sum_i a_{ik}) b_{kj} \end{aligned} \tag{14}$$

for $t \neq j$, the column sum of $C$ is

$$\sum_i c_{it} = \sum_i \sum_k a_{ik} b_{kt}$$
$$= \sum_k (\sum_i a_{ik}) b_{kt} \tag{15}$$

We first show that $j$-th column remains the lowest column sum in $C$. For $t \neq j$:

$$\sum_i c_{it} - \sum_i c_{ij} = \sum_k (\sum_i a_{ik})(b_{kt} - b_{kj})$$
$$= \sum_{k \neq j} (\sum_i a_{ik})(b_{kt} - b_{kj}) + (\sum_i a_{ij})(b_{jt} - b_{jj})$$
$$\overset{(i)}{>} \sum_{k \neq j} (\sum_i a_{ij})(b_{kt} - b_{kj}) + (\sum_i a_{ij})(b_{jt} - b_{jj})$$
$$= (\sum_i a_{ij}) \left( \sum_{k \neq j} (b_{kt} - b_{kj}) + (b_{jt} - b_{jj}) \right) \tag{16}$$
$$= \sum_i a_{ij} \left( \sum_k b_{kt} - \sum_k b_{kj} \right)$$
$$\overset{(ii)}{>} 0$$

(i) holds because for $k \neq j$, $b_{kt} - b_{kj} > 0$ and $\sum_i a_{ij} < \sum_i a_{ik}$
(ii) holds because the $j$-th column has the lowest column sum in B

We then show that $(i, j)$-th entry remains the lowest value in $i$-th row of $C$ for $i \neq j$. For $t \neq j$, we have

$$c_{it} - c_{ij} = \sum_k a_{ik} b_{kt} - \sum_k a_{ik} b_{kj}$$
$$= \sum_{k \neq j} a_{ik}(b_{kt} - b_{kj}) + a_{ij}(b_{jt} - b_{jj})$$
$$\overset{(iii)}{>} \sum_{k \neq j} a_{ij}(b_{kt} - b_{kj}) + a_{ij}(b_{jt} - b_{jj})$$
$$= a_{ij} \left( \sum_{k \neq j} (b_{kt} - b_{kj}) + (b_{jt} - b_{jj}) \right) \tag{17}$$
$$= a_{ij} \left( \sum_k b_{kt} - \sum_k b_{kj} \right)$$
$$\overset{(iv)}{>} 0$$

(iii) holds since $b_{kt} - b_{kj} > 0$ and $a_{ik} > a_{ij}$ for $i, k \neq j$.

(iv) holds because $\sum_k b_{kt} > \sum_k b_{kj}$ $\qquad\square$

For time-inhomogenous trust matrix, Assumptions 1 2 ensure the Markov chain update: $\psi_i^{(t)} = \sum_j w_{ij}^{(t)} \psi_j^{(t-1)}$, which is followed by consensus as proven in Theorem 1. We see that $b$-th column remains the lowest column sum in the product $W^{(\tau)} W^{(\tau-1)} ... W^{(1)}$, by iteratively applying Proposition 7. For $t \geq \tau$, multiplying consensus with any row stochastic preserves the consensus. Thus, the $b$-th column will remain to be the smallest column in the consensus. For the time-homogenous case, as long as $W$ holds the same properties, one can easily verify that the same result still holds. Thus, Proposition 4 is proved.

**Extend to more than one node with low-quality data.** For more than one low-quality node, what are the desired properties (sufficient conditions) for the transition (trust) matrices to have? It turns out that apart from the two conditions in a single low-quality node case, we need an extra assumption.

**Proposition 8.** *Given Assumptions 1 2 and that all agents are over-parameterized, let $\mathcal{R}$ be the set of indices of regular nodes, and $\mathcal{B}$ be the set of indices of low-quality nodes, if for $t \leq \tau$, $\mathbf{W}^{(t)}$ satisfies the following conditions:*

*i) any regular node's column sum is larger than any low-quality node's: $\min_{r \in \mathcal{R}} \sum_i w_{ir}^{(t)} > \max_{b \in \mathcal{B}} \sum_i w_{ib}^{(t)}$;*

*ii) the gap between the sum of trust from regular nodes towards any regular node $r$ and low-quality node $b$ is larger than the gap between low-quality node $b$'s self-confidence and its trust towards the regular node: $\sum_{n \in \mathcal{R}}(w_{nr}^{(t)} - w_{nb}^{(t)}) > (w_{bb}^{(t)} - w_{br}^{(t)})$,*

*iii) any node's trust towards a regular node is bigger or equal than its trust towards a low-quality node other than itself: for any $r \in \mathcal{R}$ and any $b \in \mathcal{B}$, we have $w_{nr}^{(t)} \geq w_{nb}^{(t)}$ holds as long as $n \neq b$.*

*And after $t > \tau$, $\mathbf{W}^{(t)} = \mathbf{1}\mathbf{1}^{\top}\frac{1}{N}$. Then we have nodes in $\mathcal{B}$ having a lower importance in the consensus than nodes in $\mathcal{R}$.*

*Proof.* First, let us look at the multiplication of two such matrices when $1 < t < \tau$, for any $r \in \mathcal{R}$ and $b \in \mathcal{B}$, we have conditions (1)(2)(3) remain to be true for the product $\mathbf{W}^{(t)}\mathbf{W}^{(t-1)}$. We will verify them one by one in the following part:

Verification of condition (1): any regular node's column sum is larger than any low-quality node's in $\mathbf{W}^{(t)}\mathbf{W}^{(t-1)}$. For any $r \in \mathcal{R}$ and any $b \in \mathcal{B}$, we have

$$\sum_i \sum_n w_{in}^{(t)} w_{nr}^{(t-1)} - \sum_i \sum_n w_{in}^{(t)} w_{nb}^{(t-1)}$$

$$= \sum_n (\sum_i w_{in}^{(t)}) \left(w_{nr}^{(t-1)} - w_{nb}^{(t-1)}\right)$$

$$= \sum_{n \in \mathcal{R}} (\sum_i w_{in}^{(t)}) \left(w_{nr}^{(t-1)} - w_{nb}^{(t-1)}\right) + \sum_{n \in \mathcal{B} \backslash \{b\}} (\sum_i w_{in}^{(t)}) \left(w_{nr}^{(t-1)} - w_{nb}^{(t-1)}\right)$$

$$+ (\sum_i w_{ib}^{(t)}) \left(w_{br}^{(t-1)} - w_{bb}^{(t-1)}\right)$$

$$\overset{(i)}{>} \sum_{n \in \mathcal{R}} (\sum_i w_{ib}^{(t)}) \left(w_{nr}^{(t-1)} - w_{nb}^{(t-1)}\right) + \sum_i w_{ib}^{(t)} \left(w_{br}^{(t-1)} - w_{bb}^{(t-1)}\right)$$

$$+ \sum_{n \in \mathcal{B} \backslash \{b\}} (\sum_i w_{in}^{(t)}) \left(w_{nr}^{(t-1)} - w_{nb}^{(t-1)}\right)$$

$$= (\sum_i w_{ib}^{(t)}) \left(\sum_{n \in \mathcal{R}} w_{nr}^{(t-1)} - \sum_{n \in \mathcal{R}} w_{nb}^{(t-1)} + w_{br}^{(t-1)} - w_{bb}^{(t-1)}\right)$$

$$+ \sum_{n \in \mathcal{B} \backslash \{b\}} (\sum_i w_{in}^{(t)}) \left(w_{nr}^{(t-1)} - w_{nb}^{(t-1)}\right)$$

$$\overset{(ii)}{>} 0$$

(i) holds because $\sum_i w_{in}^{(t)}$ for any $n \in \mathcal{R}$ is larger than $\sum_i w_{ib}^{(t)}$ for any $b \in \mathcal{B}$, which follows from condition (1), and $w_{nr}^{(t)} - w_{nb}^{(t)} > 0$, which follows from condition (3).

(ii) holds following the conditions (2) and (3). From (2), $\sum_{n \in \mathcal{R}} w_{nr}^{(t-1)} - \sum_{n \in \mathcal{R}} w_{nb}^{(t-1)} + w_{br}^{(t-1)} - w_{bb}^{(t-1)} > 0$, and from (3), $w_{nr}^{(t-1)} \geq w_{nb}^{(t-1)}$ for $n \neq b$

Verification of condition (2):

$$\sum_{n\in\mathcal{R}}\left(\sum_p w_{np}^{(t)}w_{pr}^{(t-1)} - \sum_p w_{np}^{(t)}w_{pb}^{(t-1)}\right) - \left(\sum_p w_{bp}^{(t)}w_{pb}^{(t-1)} - \sum_p w_{bp}^{(t)}w_{pr}^{(t-1)}\right)$$

$$=\sum_p \left(\sum_{n\in\mathcal{R}} w_{np}^{(t)} + w_{bp}^{(t)}\right)w_{pr}^{(t-1)} - \sum_p \left(\sum_{n\in\mathcal{R}} w_{np}^{(t)} + w_{bp}^{(t)}\right)w_{pb}^{(t-1)}$$

$$=\sum_p \left(\sum_{n\in\mathcal{R}} w_{np}^{(t)} + w_{bp}^{(t)}\right)\left(w_{pr}^{(t-1)} - w_{pb}^{(t-1)}\right)$$

$$=\sum_{p\in\mathcal{R}} \left(\sum_{n\in\mathcal{R}} w_{np}^{(t)} + w_{bp}^{(t)}\right)\left(w_{pr}^{(t-1)} - w_{pb}^{(t-1)}\right) + \sum_{p\in\mathcal{B}\setminus\{b\}} \left(\sum_{n\in\mathcal{R}} w_{np}^{(t)} + w_{bp}^{(t)}\right)\left(w_{pr}^{(t-1)} - w_{pb}^{(t-1)}\right)$$

$$+ \left(\sum_{n\in\mathcal{R}} w_{nb}^{(t)} + w_{bb}^{(t)}\right)\left(w_{br}^{(t-1)} - w_{bb}^{(t-1)}\right)$$

$$\overset{(iii)}{\geq} \sum_{p\in\mathcal{R}} \left(\sum_{n\in\mathcal{R}} w_{nb}^{(t)} + w_{bb}^{(t)}\right)\left(w_{pr}^{(t-1)} - w_{pb}^{(t-1)}\right) + \left(\sum_{n\in\mathcal{R}} w_{nb}^{(t)} + w_{bb}^{(t)}\right)\left(w_{br}^{(t-1)} - w_{bb}^{(t-1)}\right)$$

$$+ \sum_{p\in\mathcal{B}\setminus\{b\}} \left(\sum_{n\in\mathcal{R}} w_{np}^{(t)} + w_{bp}^{(t)}\right)\left(w_{pr}^{(t-1)} - w_{pb}^{(t-1)}\right)$$

$$=\left(\sum_{n\in\mathcal{R}} w_{nb}^{(t)} + w_{bb}^{(t)}\right)\left(\sum_{p\in\mathcal{R}} w_{pr}^{(t-1)} - \sum_{p\in\mathcal{R}} w_{pb}^{(t-1)} + w_{br}^{(t-1)} - w_{bb}^{(t-1)}\right)$$

$$+ \sum_{p\in\mathcal{B}\setminus\{b\}} \left(\sum_{n\in\mathcal{R}} w_{np}^{(t)} + w_{bp}^{(t)}\right)\left(w_{pr}^{(t-1)} - w_{pb}^{(t-1)}\right)$$

$$\overset{(iv)}{\geq} 0$$

(iii) holds because for $p$ a regular node, we have $\sum_{n\in\mathcal{R}} w_{np}^{(t)} + w_{bp}^{(t)} > \sum_{n\in\mathcal{R}} w_{nb}^{(t)} + w_{bb}^{(t)}$, which follows from condition (2), and $w_{pr}^{(t-1)} - w_{pb}^{(t-1)} \geq 0$ for $p \neq b$, following from condition (3).
(iv) holds because of conditions (2) and (3).

Verification of (3): for $n \neq b$, we want to show the trust towards a regular node $r$ is bigger than towards a low-quality node $b$, that is $\sum_p w_{np}^{(t)}w_{pr}^{(t)} > \sum_p w_{np}^{(t)}w_{pb}^{(t)}$

$$\sum_p w_{np}^{(t)}w_{pr}^{(t)} - \sum_p w_{np}^{(t)}w_{pb}^{(t)}$$

$$=\sum_{p\in\mathcal{R}} w_{np}^{(t)}\left(w_{pr}^{(t)} - w_{pb}^{(t)}\right) + \sum_{p\in\mathcal{B}\setminus\{b\}} w_{np}^{(t)}\left(w_{pr}^{(t)} - w_{pb}^{(t)}\right) + w_{nb}^{(t)}\left(w_{br}^{(t)} - w_{bb}^{(t)}\right)$$

$$\overset{(v)}{\geq} \sum_{p\in\mathcal{R}} w_{nb}^{(t)}\left(w_{pr}^{(t)} - w_{pb}^{(t)}\right) + w_{nb}^{(t)}\left(w_{br}^{(t)} - w_{bb}^{(t)}\right) + \sum_{p\in\mathcal{B}\setminus\{b\}} w_{np}^{(t)}\left(w_{pr}^{(t)} - w_{pb}^{(t)}\right)$$

$$=w_{nb}^{(t)}\left(\sum_{p\in\mathcal{R}} w_{pr}^{(t)} - \sum_{p\in\mathcal{R}} w_{pb}^{(t)} + w_{br}^{(t)} - w_{bb}^{(t)}\right) + + \sum_{p\in\mathcal{B}\setminus\{b\}} w_{np}^{(t)}\left(w_{pr}^{(t)} - w_{pb}^{(t)}\right)$$

$$\overset{(vi)}{\geq} 0$$

(v) holds because for $n \neq b$, we have $w_{np}^{(t)} \geq w_{nb}^{(t)}$, following from condition (3), and $w_{pr}^{(t)} - w_{pb}^{(t)} \geq 0$ for $p \neq b$.

(vi) holds following from conditions (2) and (3).

It follows that in the product $W^{(\tau)}W^{(\tau-1)}...W^{(1)}$, a low-quality node will still have a lower column sum than any regular node. Because conditions (1)(2)(3) holds for any product of $W^{(t)}$'s as long as each of the $W^{(t)}$ share the conditions listed by (1)(2)(3).

After $t > \tau$, multiplying with a naive weight matrix does not change the column sum order, we will have all low-quality nodes have lower importance in the consensus than the regular nodes.

$\square$

# F    Reasoning for confidence weighting factor $\beta_i^{(t)}$

In this section, we justify our choice of $\beta_i^{(t)}(x)$ in Section 4.3, i.e. we show via adding such a term, we are able to downweight a regular node's trust towards a bad node.
$\Phi^{(t)}$ is a row-normalized pairwise cosine similarity matrix, with $(i,j)$-th entry *before* row normalization as

$$\frac{1}{n^\star} \sum_{x' \in X^*} \frac{\left\langle f_{\theta_i^{(t-1)}}(x'), f_{\theta_j^{(t-1)}}(x') \right\rangle}{\|f_{\theta_i^{(t-1)}}(x')\|_2 \|f_{\theta_j^{(t-1)}}(x')\|_2} \tag{18}$$

After adding a $\beta_i^{(t)}(x) = 1/\mathcal{H}(f_{\theta_i^{(t-1)}}(x))$, we have $W^{(t)}$ with $(i,j)$-th entry *before* row normalization as

$$\frac{1}{n^\star} \sum_{x' \in X^*} \frac{1}{\mathcal{H}(f_{\theta_i^{(t-1)}}(x'))} \frac{\left\langle f_{\theta_i^{(t-1)}}(x'), f_{\theta_j^{(t-1)}}(x') \right\rangle}{\|f_{\theta_i^{(t-1)}}(x')\|_2 \|f_{\theta_j^{(t-1)}}(x')\|_2} \tag{19}$$

We want to show that the weighting scheme down-weights a regular node $i$'s trust towards a low-quality node $b$, that is

$$\phi_{ib}^{(t)} > w_{ib}^{(t)}$$

As the comparison is made with respect to the same time step $t$, we drop the $t$ notation from now on. Let $\{a_0, .., a_{N-1}\}$ be the cosine similarity between a regular agent $i$ and others inside agent $i$'s confident region, and $\{b_0, .., b_{N-1}\}$ be the cosine similarity between $i$ and others outside agent $i$'s confident region. By confident region, we mean region with low entropy in class probabilities, i.e. the model is more sure about the prediction. Further, we make the following assumptions:

a) for $x'$ in agent $i$'s confident region, we have low entropy of predicted class probabilities: $\mathcal{H}(f_{\theta_i^{(t-1)}}(x')) = 1/c_1$; while for $x'$ outside agent $i$'s confident region, we have $\mathcal{H}(f_{\theta_i^{(t-1)}}(x')) = 1/c_2$. We further assume $0 < c_2 < c_1$.

b) inside a regular node $i$'s confident region, $i$ has a better judgment of the alignment score produced by cosine similarity, such that the cosine similarity with low quality $b$ is weighted lower inside:

$$\frac{a_b}{\sum_j a_j} < \frac{b_b}{\sum_j b_j} \tag{20}$$

to claim $w_{ib} < \phi_{ib}$, we need to show

$$\frac{c_1 a_b + c_2 b_b}{\sum_j (c_1 a_j + c_2 b_j)} < \frac{a_b + b_b}{\sum_j (a_j + b_j)} \tag{21}$$

*Proof.* Re-arrange Equation 20, we get

$$b_b \sum_j a_j > a_b \sum_j b_j \tag{22}$$

Multiply with $c_2 - c_1$ on both sides, we have

$$(c_2 - c_1) b_b \sum_j a_j < (c_2 - c_1) a_b \sum_j b_j \tag{23}$$

$$c_2 b_b \sum_j a_j + c_1 a_b \sum_j b_j \; < c_1 b_b \sum_j a_j \; + c_2 a_b \sum_j b_j \tag{24}$$

Now add $c_1 a_b \sum_j a_j + c_2 b_b \sum_j b_j$ to both sides, we have

$$c_1 a_b \sum_j a_j + c_2 b_b \sum_j a_j + c_1 a_b \sum_j b_j + c_2 b_b \sum_j b_j <$$
$$c_1 a_b \sum_j a_j + c_1 b_b \sum_j a_j \; + c_2 a_b \sum_j b_j + c_2 b_b \sum_j b_j \tag{25}$$

Combining the terms we have

$$(c_1 a_b + c_2 b_b) \left( \sum_j (a_j + b_j) \right) < \left( \sum_j (c_1 a_j + c_2 b_j) \right) (a_b + b_b) \tag{26}$$

following which, we directly have

$$\frac{c_1 a_b + c_2 b_b}{\sum_j (c_1 a_j + c_2 b_j)} < \frac{a_b + b_b}{\sum_j (a_j + b_j)} \tag{27}$$

$\square$

# G    Complementary details

## G.1    Details regarding model training

All the model training was done using a single GPU (NVIDIA Tesla V100). For each local iteration, we load local data and shared unlabeled data with batch size 64 and 256 separately. We empirically observed that a larger batch size for unlabeled data is necessary for the training to work well. The optimizer used is Adam with a learning rate 5e-3. For Cifar10 and Cifar100, as the base model is not pretrained, we do 50 global rounds with 5 local training epochs for each agent per global round. For Fed-ISIC-2019 dataset, as the base model is pretrained EfficientNet, we do 20 global rounds. For the first 5 global rounds, we set $\lambda = 0$ to arrive at good local models, such that every agent can evaluate trust more fairly. After that, $\lambda$ is fixed as 0.5. *Dynamic* trust is computed after each global round, while *static* trust denotes the utilization of the initially calculated trust value throughout the whole experiment.

For Cifar10 and Cifar100, we use 5% of the whole dataset to constitute $\boldsymbol{X}^*$, where each class has equal representation. For the rest, we spread them into 10 clients using Dirichlet distribution with $\alpha = 1$. For Fed-ISIC-2019 dataset, we follow the original splits as in du Terrail et al. [34], and we let each client contribute 50 data samples to constitute $\boldsymbol{X}^*$.

We employ a fixed $\lambda$ for all our experiments. To select $\lambda$, we randomly sample 10% of the full Cifar10 dataset, which we then split into local training data (95%) and $\boldsymbol{X}^*$ (5%). The local training data is then spread into 10 clients using Dirichlet distribution with $\alpha = 1$. The test global accuracy and value of $\lambda$ is plotted out in Fig. 8. We thus choose $\lambda = 0.5$ for all our experiments, and it is always able to give stable performances according to our experiments.

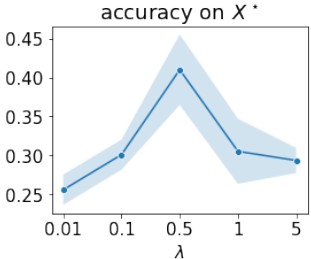

Figure 8: $\lambda$ versus algorithm performance

## G.2    Limitations of the work

The main limitation of this work is the requirement of an extra shared unlabelled dataset, like in other knowledge distillation-based decentralized learning works. Moreover, each agent needs to calculate their trust towards all other nodes locally. The extra computational complexity is $\mathcal{O}(N^2 \times n^\star \times C)$ per global round, where $N$ stands for the number of agents, $n^\star$ stands for the size of the shared dataset and $C$ denotes the number of classes. The computation can be heavy if the number of clients gets large. But as we focus on cross-silo setting, $N$ usually does not tend to be a big number.

