# 7  Appendix

## 7.1  Polynomial regression examples

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

## 7.6 Reasoning for confidence upweighting block

In this section, we provide our intuition of adding such a confidence weighting block in Equation (7).

$\mathbf{\Phi}^{(t)}$ is a row-normalized pairwise cosine similarity matrix, with $(i,j)$-th entry before row normalization as

$$\frac{1}{n_S} \sum_{\boldsymbol{x}' \in \boldsymbol{X}_s} \frac{\left\langle \boldsymbol{f}_{\boldsymbol{\theta}_i^{(t-1)}}(\boldsymbol{x}'), \boldsymbol{f}_{\boldsymbol{\theta}_j^{(t-1)}}(\boldsymbol{x}') \right\rangle}{\|\boldsymbol{f}_{\boldsymbol{\theta}_i^{(t-1)}}(\boldsymbol{x}')\|_2 \|\boldsymbol{f}_{\boldsymbol{\theta}_j^{(t-1)}}(\boldsymbol{x}')\|_2} \tag{19}$$

After adding a confidence weighting block, we have $\boldsymbol{W}^{(t)}$ with $(i,j)$-th entry before row normalization as

$$\frac{1}{n_S} \sum_{\boldsymbol{x}' \in \boldsymbol{X}_s} \frac{1}{\mathcal{H}(\boldsymbol{f}_{\boldsymbol{\theta}_i^{(t-1)}}(\boldsymbol{x}'))} \frac{\left\langle \boldsymbol{f}_{\boldsymbol{\theta}_i^{(t-1)}}(\boldsymbol{x}'), \boldsymbol{f}_{\boldsymbol{\theta}_j^{(t-1)}}(\boldsymbol{x}') \right\rangle}{\|\boldsymbol{f}_{\boldsymbol{\theta}_i^{(t-1)}}(\boldsymbol{x}')\|_2 \|\boldsymbol{f}_{\boldsymbol{\theta}_j^{(t-1)}}(\boldsymbol{x}')\|_2} \tag{20}$$

We want to show that the weighting scheme down-weights the a regular node $i$'s trust towards a low-quality node $b$, that is

$$\phi_{ib}^{(t)} > w_{ib}^{(t)}$$

As the comparison is made with respect to the same time step $t$, we drop the $t$ notation from now on. Let $\{a_0, .., a_{N-1}\}$ be the cosine similarity between a regular agent $i$ and others inside agent $i$'s confident region, and $\{b_0, .., b_{N-1}\}$ be the cosine similarity between $i$ and others outside agent $i$'s confident region. By confident region, we mean region with low entropy in class probabilities, i.e. the model is more sure about the prediction. Further, we make the following assumptions:

(1) for $\boldsymbol{x}'$ in agent $i$'s confident region, we have low entropy of predicted class probabilities: $\mathcal{H}(\boldsymbol{f}_{\boldsymbol{\theta}_i^{(t-1)}}(\boldsymbol{x}')) = 1/c_1$ with $c_1 > 1$, while for $\boldsymbol{x}'$ outside agent $i$'s confident region, we have $\mathcal{H}(\boldsymbol{f}_{\boldsymbol{\theta}_i^{(t-1)}}(\boldsymbol{x}')) = 1/c_2$ with $c_2 < 1$

(2) inside a regular node $i$'s confident region, $i$ has a better judgment of the alignment score produced by cosine similarity, such that the cosine similarity with low quality $b$ is weighted lower inside:

$$\frac{a_b}{\sum_i a_i} < \frac{b_b}{\sum_i b_i} \tag{21}$$

to claim $w_{ib} < \phi_{ib}$, we need to show

$$\frac{c_1 a_b + c_2 b_b}{\sum_i (c_1 a_i + c_2 b_i)} < \frac{a_b + b_b}{\sum_i (a_i + b_i)} \tag{22}$$

*Proof.* Re-arrange Equation 21, we get

$$b_b \sum_i a_i > a_b \sum_i b_i \tag{23}$$

Multiply with $c_2 - c_1$ on both sides, we have

$$(c_2 - c_1) b_b \sum_i a_i < (c_2 - c_1) a_b \sum_i b_i \tag{24}$$

$$c_2 b_b \sum_i a_i + c_1 a_b \sum_i b_i < c_1 b_b \sum_i a_i + c_2 a_b \sum_i b_i \tag{25}$$

Now add $c_1 a_b \sum_i b_i + c_2 b_b \sum_i b_i$ to both sides, we have

$$c_1 a_b \sum_i a_i + c_2 b_b \sum_i a_i + c_1 a_b \sum_i b_i + c_2 b_b \sum_i b_i <$$
$$c_1 a_b \sum_i a_i + c_1 b_b \sum_i a_i + c_2 a_b \sum_i b_i + c_2 b_b \sum_i b_i \tag{26}$$

673 Combining the terms we have

$$(c_1 a_b + c_2 b_b) \left( \sum_i (a_i + b_i) \right) < \left( \sum_i (c_1 a_i + c_2 b_i) \right) (a_b + b_b) \tag{27}$$

674 following which, we directly have

$$\frac{c_1 a_b + c_2 b_b}{\sum_i (c_1 a_i + c_2 b_i)} < \frac{a_b + b_b}{\sum_i (a_i + b_i)} \tag{28}$$

675 $\square$

### 7.7 Complementary details

#### 7.7.1 Details regarding model training

678 All the model training was done using a single GPU (NVIDIA Tesla V100). For each local iteration,
679 we load local data and shared unlabeled data with batch size 64 and 256 separately. We empirically
680 observed that a larger batch size for unlabeled data is necessary for the training to work well. The
681 optimizer used is Adam with a learning rate 5e-3. For Cifar10 and Cifar100, as the base model is not
682 pretrained, we do 50 global rounds with 5 local training epochs for each agent per global round. For
683 Fed-ISIC-2019 dataset, as the base model is pretrained EfficientNet, we do 20 global rounds. For the
684 first 5 global rounds, we set $\lambda = 0$ to arrive at good local models, such that every agent can evaluate
685 trust more fairly. After that, $\lambda$ is fixed as 0.5. *Dynamic* trust is computed after each global round,
686 while *static* trust denotes the utilization of the initially calculated trust value throughout the whole
687 experiment.

688 For Cifar10 and Cifar100, we use 5% of the whole dataset to constitute $\boldsymbol{X_s}$, where each class has
689 equal representation. For the rest, we spread them into 10 clients using Dirichlet distribution with
690 $\alpha = 1$. For Fed-ISIC-2019 dataset, we follow the original splits as in du Terrail et al. [33], and we let
691 each client contribute 50 data samples to constitute $\boldsymbol{X_s}$.

692 We employ a fixed $\lambda$ for all our experiments. To select $\lambda$, we randomly sample 10% of the full
693 Cifar10 dataset, which we then split into local training data (95%) and $\boldsymbol{X_s}$ (5%). The local training
694 data is then spread into 10 clients using Dirichlet distribution with $\alpha = 1$. The test global accuracy
695 and value of $\lambda$ is plotted out in Figure 8. We thus choose $\lambda = 0.5$ for all our experiments, and it is
696 always able to give stable performances according to our experiments.

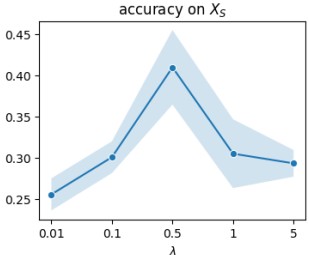

Figure 8: $\lambda$ versus algorithm performance

#### 7.7.2 Limitations of the work

698 The main limitation of this work is the requirement of an extra shared unlabelled dataset, like in other
699 knowledge distillation-based decentralized learning works. Moreover, each agent needs to calculate
700 their trust towards all other nodes locally. The extra computational complexity is $\mathcal{O}(N \times n_S \times C)$,
701 where $N$ stands for the number of agents, $n_S$ stands for the size of the shared dataset and $C$ denotes
702 the number of classes. The computation can be heavy if the number of clients gets large. But as we
703 focus on cross-silo setting, $N$ usually does not tend to be a big number.