# OpenReview forum: "Collaborative Learning via Prediction Consensus"
_NeurIPS.cc/2023/Conference — NeurIPS 2023 poster_

### Official Review · Reviewer_Bri9 · 2023-07-05

**Soundness:** 3 good
**Presentation:** 3 good
**Contribution:** 3 good
**Rating:** 5
**Confidence:** 3

**Summary:**

To facilitate information exchange among agents and improve prediction accuracy on a shared target domain, this paper proposes a decentralized learning algorithm based on prediction consensus inspired from social science for leveraging each agent’s predictions. To be concrete, each agent firstly shares its prediction and weighs these predictions according to the trust scores, then these proxy labels are used to augment local model training via distillation. The authors show the trust measure based on cosine similarity can attain trust matrix with ideal properties, which could facilitate effective consensus. The efficacy is empirically demonstrated by showing that the proposed method is better than classical baselines under some heterogeneous conditions.

**Strengths:**

This paper considers a collaborative learning setting where each agent owns their own data and agents want to collaborate to improve their predictive performance over a target domain. The authors propose a decentralized algorithm based on distillation and trust weighting scheme, which is useful in cases where data sharing and model sharing are not allowed.  The key ingredient, which is called trust, is inspired from social science, which is an interesting topic. The theoretical analysis seems sound although I didn’t check all the details. The authors also conduct sufficient experiments to demonstrate the effectiveness of the proposed method. Overall, the paper is a well-written.

**Weaknesses:**

In the collaborative setting, each agent owns their own data and wants to improve its predictive accuracy while keeping its data and model private, which is a significant problem. However, exchanging predictions may leak valuable information. Is there a more private way to share data between agents?
The goal of agents is to improve the prediction performance on a shared target domain.  Is it possible that ensemble algorithms like bagging work better here?  I think the motivation and necessity of collective prediction could be improved.
I didn’t clearly figure out why the proposed method could keep communication at a minimum as claimed in the paper. Could the authors give more detailed explanations?
The explanation of co-training in line 144 is not that convincing. As far as I know, dispersed and heterogeneous data on multiple agents cannot be simply viewed as multi-view data. Could the author give more explanations?


**Questions:**

please see weaknesses

---

> ### Author Rebuttal · Authors · 2023-08-07
>
> Thanks for your helpful comments and feedback. Regarding the weakness pointed out by the reviewer, we offer the subsequent clarifications:
>
> **W1 Information leakage**: This is an interesting question. As you point out, exchanging predictions may still leak sensitive information about the model’s training data. However, compared to the conventional ways in collaborative learning, such as sharing model parameters or gradients, predictions contain much less information. Further our scheme, due to the information exchange through model queries, renders itself amenable to applying standard query perturbation approaches to guarantee differential privacy. Crucially, the number of queries each agent is willing to answer is entirely in their own control, which poses a significant advantage. We will add a discussion of privacy considerations. In the worst case privacy constraints would limit the number of outer rounds that can be performed (we further attached a plot of the number of outer rounds versus accuracy in the **global response PDF**). Studying this tradeoff between privacy and the progress we can make through information exchange more formally could be an interesting extension for future work.
>
> **W2  Comparison to other ensemble algorithms**: Ensemble algorithms like bagging would put equal weights on each agent’s predictions, which is equivalent to our naive trust scheme. This equal weighting will suffer from low-quality agents that possess weak models or bad-quality data, please refer to Figure 2 and Table 1 for supporting experimental results. This also motivates the design of our collaborative learning algorithm that is able to benefit from each agent’s local expertise and can learn to upweight high-quality agents and downweight low-quality agents in the consensus.
>
> **W3  Minimum communication costs**: Regarding our claim that “the proposed method could keep communication at a minimum”, we position ourselves in the modern deep learning regime where the size of model parameters comes in millions or billions. Compared to exchange model parameters/gradients, which induces communication complexity per global round $\mathcal{O}(N \times |params|)$, exchanging model predictions will induce communication cost $\mathcal{O}(N^2 \times n_S \times C)$, where $N$ stands for the number of agents, $n_S$ stands for the size of the shared dataset and $C$ denotes the number of classes. The latter is significantly smaller. We will make the meaning of the claim more precise in this sense.
>
> **W4 Multi-view explanation**: Thanks for pointing this out. In our setting, we are not talking about combining models with different feature views. Instead, each agent contributes their knowledge covering different regions of the target domain, and thus combining their knowledge wisely leads to a better model over the full region of the target domain. We will make sure this is made more explicit.

---

> > ### Comment · Reviewer_Bri9 · 2023-08-18
> >
> > Thanks for the authors' response. I would like to retain my evaluation and keep my score.

---

### Official Review · Reviewer_TUxA · 2023-07-06

**Soundness:** 3 good
**Presentation:** 2 fair
**Contribution:** 3 good
**Rating:** 6
**Confidence:** 4

**Summary:**

This paper proposes a collaborative learning method that leverages unlabeled auxiliary data to facilitate the exchange of expertise among agents. The method adaptively weights the influence of each collaborator on the pseudo-labels until a consensus on how to label the auxiliary data is reached. The authors demonstrate that their collaboration scheme significantly boosts individual model's performance with respect to the global distribution, compared to local training. They also show that their method is particularly effective in scenarios where the intrinsic beliefs of individuals counterbalance the averaging process and yield a diversity of opinions.

**Strengths:**

1. The proposed approach utilizes unlabeled auxiliary data to enhance the exchange of expertise among agents, resulting in a significant improvement in individual model performance when compared to local training with respect to the global distribution.
2. The method dynamically assigns weights to each collaborator's influence on the pseudo-labels, iteratively reaching a consensus on how to label the auxiliary data. This adaptive weighting effectively detects and mitigates the negative impact of poor models on the collective performance.
3. The authors demonstrate the method's efficacy, particularly in scenarios where individuals hold diverse beliefs that counterbalance the averaging process. This diversity of opinions leads to improved performance in heterogeneous environments, showcasing the method's potential in such situations.

**Weaknesses:**

1. The paper lacks a comprehensive comparison with other recent state-of-the-art collaborative learning methods, making it challenging to evaluate the relative performance of the proposed method. Including such comparisons would enhance the understanding of its strengths and weaknesses.
2. The paper would benefit from a more in-depth analysis of the computational complexity of the proposed method. This analysis would shed light on its scalability to large-scale datasets or complex models, which is crucial for practical implementation.
3. The paper assumes the trustworthiness and non-malicious nature of all agents, which may not be realistic in real-world scenarios. To address this limitation, a more robust trust weighting scheme capable of handling malicious agents should be considered, ensuring the method's applicability in diverse environments.

**Questions:**

1. Can you provide more details on the computational complexity of the proposed method? How does it scale to large-scale datasets or complex models?
2. How does the proposed method compare to other state-of-the-art collaborative learning methods in terms of performance? Can you provide a comprehensive comparison?
3. How robust is the trust weighting scheme to malicious agents? Have you considered scenarios where some agents may intentionally provide incorrect information?
4. Have you tested the proposed method on real-world datasets? If so, can you provide some examples and discuss the results?
5. How sensitive is the proposed method to the choice of hyperparameters? Have you conducted a sensitivity analysis to assess the impact of different hyperparameters on performance?

**Limitations:**

The paper briefly mentions some potential limitations of the proposed method, such as the lack of a comprehensive comparison with other state-of-the-art methods and the assumption of trustworthy and non-malicious agents. However, the authors do not provide a detailed discussion of these limitations or potential negative impacts of their work. While the paper does provide some insights into the strengths and weaknesses of the proposed method, a more thorough analysis of the limitations and potential negative impacts would be desirable.

---

> ### Author Rebuttal · Authors · 2023-08-07
>
> Thanks for your positive feedback and insightful questions. We address the raised questions below:
>
>
> **Q1**: Compared to typical decentralized algorithms where model params/gradients are communicated, our method introduced extra computational complexity on the calculation of pairwise trust scores and the computation brought by including the augmented dataset into training. The first one is matrix calculation which can be made efficient on GPU, and the extra computing time mainly comes from the augmented dataset, which is 1.4x longer in our experiments with one V100 GPU.
>
>
> However, we want to emphasize that our method _greatly reduced communication complexity_, which is the main bottleneck in decentralized training. “In federated optimization communication costs dominate — we will typically be limited by an upload bandwidth of 1 MB/s or less” [3]. Conventional collaborative learning algorithm communicates model params/gradients, which come at a size of millions or billions, and the communication complexity per global round is $\mathcal{O}(N \times |params|)$. While we communicate model predictions. The complexity is $\mathcal{O}(n_S \times C \times N^2)$ for each global round, where $N$ stands for the number of agents, $n_S$ stands for the size of the shared dataset and $C$ denotes the number of classes. It is clear that this value does not scale up with more complex models, and is much smaller than the model size.
>
>
> **Q2**: Based on your comment we further included two SOTA (and classic) methods that were designed to address statistical heterogeneity in federated learning: FedDyn [R1] and SCAFFOLD [R2]. SCAFFOLD uses variance reduction to correct for the `client-drift' in its local updates, and FedDyn designs a dynamic regularization term to ensure the alignment of global and device solutions. In our first scenario where the same model architecture is applied, we did a more comprehensive comparison. Please note that the training loss of SCAFFOLD and FedDyn gets saturated when the number of local epochs is set to 5, which is our default setting. Instead, we did a quick hyperparameter search, and found the optimal numbers of local epochs for SCAFFOLD and FedDyn are 1 and 2 respectively, and report the corresponding accuracy on our target dataset. For all the other methods, we set the number of local epochs to 5 as reported in the paper. Our proposed method still occupies the top 1 accuracy in most of the cases. Please refer to Table 1 in the **global response PDF file**.
>
> **Q3**: In our paper we design the algorithm under the assumption that every agent communicates honestly, that is, no Byzantine workers that send intentionally incorrect information are involved. However, our method does exhibit some robustness against a typical Byzantine attack, which is label flip (we call workers with flipped labels low-quality workers in the paper). With 2 out of 10 workers having 100% flipped labels, we did not witness a big performance drop.
>
> Additionally, here are some of our thoughts regarding your question: If malicious workers that send intentionally incorrect information are involved, then the nodes might reject to reach consensus, instead of reaching a detrimental bad consensus, assuming a reasonable $\lambda$ is chosen. Imagine if a bad consensus is reached with malicious nodes involved, then for the regular nodes, the consensus loss and the local loss will not decrease in the same direction, and thus the consensus solution is not a stationary solution. Indeed, the "personal" part of our loss adds some degree of robustness to malicious nodes.
>
> We agree that it would be interesting to investigate robustness against _intentionally_ malicious nodes more.  However, we would like to keep the focus of our work on the new protocol of information exchange, and leave robustness extensions for future work.
>
>
> **Q4**: Yes, Fed-ISIC-2019 [30, 31, 32] is a real-world dataset from the healthcare domain. It is a dermoscopic dataset collected from 6 different hospitals, which is a benchmark dataset from Flamby paper [33]. Our method shows consistent success in the classification task. Please further refer to Figure 1 and Figure 3(c) for more details.
>
> **Q5**: We offer sensitivity analysis of how the performance scales with $\lambda$ and the number of local epochs. Accuracy in the target domain versus different choices of hyperparameter $\lambda$ is plotted in Figure 8 in the appendix. And the influence of the number of local epochs is also presented in Figure 6 in the main text. Is there another study you would be interested in?
>
>
> [R1] Durmus Alp Emre Acar, Yue Zhao, Ramon Matas Navarro, Matthew Mattina, Paul N. Whatmough, Venkatesh Saligrama. _Federated Learning Based on Dynamic Regularization_. ICLR 2021
>
> [R2] Sai Praneeth Karimireddy, Satyen Kale, Mehryar Mohri, Sashank J. Reddi, Sebastian U. Stich, Ananda Theertha Suresh. _SCAFFOLD: Stochastic Controlled Averaging for Federated Learning_. ICML 2020

---

> > ### Comment · Reviewer_TUxA · 2023-08-17
> >
> > I recommend that the author incorporate this section into the main body of the revised paper. While I am inclined to give a higher score based on this addition, I believe it's crucial to consider the feedback from other reviewers as well. I have no further questions.

---

### Official Review · Reviewer_u6rQ · 2023-07-10

**Soundness:** 3 good
**Presentation:** 3 good
**Contribution:** 3 good
**Rating:** 7
**Confidence:** 2

**Summary:**

The paper proposes a collaborative learning approach that leverages unlabeled auxiliary data to improve individual models through consensus. The trust weighting scheme adapts to each collaborator's influence, leading to a consensus on how to label the auxiliary data. The authors demonstrate that this collaboration scheme significantly boosts individual model performance and effectively mitigates the negative impact of bad models on the collective. Overall, the paper makes a valuable contribution to the field of collaborative learning and presents a promising approach for improving model performance through consensus.

**Strengths:**

- The paper proposes a novel approach for collaborative learning through consensus on unlabeled auxiliary data.
- The collaboration scheme significantly boosts individual model performance and mitigates the negative impact of bad models on the collective.
- The paper provides a thorough description of the algorithm and its implementation.
- The experimental results demonstrate the effectiveness of the proposed approach.

**Weaknesses:**

- The paper could benefit from a more detailed analysis of the impact of different trust weighting schemes on the consensus.
- The paper could benefit from a more detailed discussion of the assumptions and limitations of the trust-based iterative pseudo-labeling process.
- The paper could provide more insights into the computational complexity of the proposed algorithm and potential scalability issues.

**Questions:**

- How does the proposed approach scale to larger datasets and more complex model architectures, and can the authors provide more insights into the computational complexity of the algorithm?

**Limitations:**

Yes.

---

> ### Author Rebuttal · Authors · 2023-08-08
>
> Thanks for your positive and helpful feedback.
> We address your question regarding computational complexity of our method in the following:
>
>
> Compared to typical decentralized algorithms where model params/gradients are communicated, our method introduced extra computational complexity on the calculation of pairwise trust scores and the computation brought by including the augmented dataset ($X_S$ and the corresponding pseudolabels) into training. The first one is matrix calculation and can be made efficient on GPU, and the extra computing time mainly comes from the augmented dataset, which is 1.35x longer per global round in our experiments with one V100 GPU. We believe this would less be an issue with more computing resources.
>
>
> However, we want to emphasize that our method greatly reduced communication complexity, which is a bottleneck in decentralized training. “In federated optimization communication costs dominate — we will typically be limited by an upload bandwidth of 1 MB/s or less” [3]. Conventional collaborative learning algorithm communicates model params/gradients, which come at size of millions, and the communication complexity per global round is $\mathcal{O}(N \times |params|)$. While we communicate model predictions. The complexity is $\mathcal{O}(n_S \times C \times N^2)$ for each global round, where $N$ stands for the number of agents, $n_S$ stands for the size of the shared dataset and $C$ denotes the number of classes. It is clear that this value does not scale up with more complex models, and is much smaller than the model size.

---

### Official Review · Reviewer_vDF7 · 2023-07-27

**Soundness:** 3 good
**Presentation:** 4 excellent
**Contribution:** 3 good
**Rating:** 5
**Confidence:** 3

**Summary:**

In this paper, the authors consider a collaborative learning setting where agents want to improve their predictive performance on a shared target domain. The paper proposes a novel algorithm based on prediction-consensus, which effectively addresses statistical and model heterogeneity in the learning process. The algorithm works by having agents pseudo-label data from the target domain. The pseudo-labels are then used to compute a trust weighting scheme, which determines how much each agent's opinion should be weighted when reaching a consensus on how to label the unlabeled data. Theoretical results that show consensus can be reached via the algorithm and justify the conditions for good consensus to be achieved. Overall, the paper is a significant contribution to the field of collaborative learning. The proposed algorithm is a promising approach for improving the predictive performance of individual models in the presence of heterogeneity.


**Strengths:**

* The paper proposes a novel algorithm for collaborative learning that is based on prediction-consensus.
* The paper is well-written and easy to follow. The authors do a good job of explaining the motivation for the work, the proposed algorithm, and the experimental results.
* The theoretical results in the paper are sound. The authors provide a rigorous analysis of the proposed algorithm and show that it can reach consensus under certain conditions.


**Weaknesses:**

* The experimental results in the paper are somewhat weak. The authors only report results on a few datasets and with a few model architectures. And the datasets seem over-simplified. It would be helpful to see more experimental results on a wider variety of datasets and with a wider variety of model architectures. This would help to give a better sense of the generalizability of the results.
* Again on the experimental results, it would be great to study the effects of the hyperparameter lambda and distance function D. It would be helpful to see how the performance of the algorithm is affected by different hyperparameter settings.

**Questions:**

* Assumption 1 states that there is no concept shift between the local data distributions. However, it seems to be a quite strong assumption. If there is no concept shift, then the problem of collaborative learning becomes much easier.
* Depending on the choice of lambda, is it possible that the algorithm converges to a point where although the local models reach consensus, the consensus is not ideal?


**Limitations:**

The authors have adequately addressed the limitations and potential negative societal impact of their work.

---

> ### Author Rebuttal · Authors · 2023-08-07
>
> Thank you for providing helpful feedback, which we genuinely appreciate.
>
> Regarding the comment concerning the over-simplicity of our chosen datasets, it is important to highlight that we have, in fact, incorporated a challenging dataset known as Fed-ISIC-2019 [30, 31, 32] from a real-world use case. This dataset comprises dermoscopic images collected from six different hospitals, thereby adding complexity to the experiments. Additionally, we employed the Cifar10/100 datasets with Dirichlet distributed splits, which are standard in collaborative learning experiments. The consistent performance observed on these datasets serves to demonstrate the potential practical benefits of our algorithm.
>
> **Q1**: In the collaborative learning setting, statistical heterogeneity remains an issue even without concept shift [5,6]. This arises due to the non-IID nature of data distributed across agents, causing each agent's local objective to deviate from the global one. Consequently, the averaged federated model may deviate from the global optima. Our algorithm effectively addresses scenarios with highly non-IID data. Furthermore, beyond concept shift, the presence of low-quality nodes (nodes with bad-quality data or weak model architectures) can impede the learning process, and our algorithm is specifically designed to address this concern.
>
> **Q2**: Yes, one could imagine with a large enough $\lambda$, such that every agent’s goal is just to reach consensus, then any consensus could be a minimizer to the optimization problem. We showed a plot of accuracy on $X_S$ versus the choice of $\lambda$ in the appendix (see Figure 8). We found that accuracy increases and then decreases again as $\lambda$ is increased and the optimal value is around 0.5. We use this value for our experiment without further hyperparameter tuning.

---

### Author Rebuttal · Authors · 2023-08-08

We thank all the reviewers for their time and effort spent in giving comments and feedbacks on the paper, and the ACs for the help in the reviewing process.

In addition to the separate responses, we further added a table and a graph in our global response PDF to better support our arguments experimentally. We kindly request your consideration of these additions for enhanced clarity and understanding.

---

### Decision · Program_Chairs · 2023-09-21

**Decision:**

Accept (poster)

**Comment:**

The paper studies collaborative learning via prediction-consensus on auxiliary data. All the reviewers appreciate the novelty of the work and  the theoretical results. Reviewers asked for stronger empirical comparisons to state-of-the-art, more in-depth discussion on computational complexity, potential privacy concerns. Please consider address them in your manuscript.